# Neural Flow Diffusion Models: Learnable Forward Process for Improved Diffusion Modelling

**Grigory Bartosh**
University of Amsterdam
g.bartosh@uva.nl

**Dmitry Vetrov**
Constructor University, Bremen
dvetrov@constructor.university

**Christian A. Naesseth**
University of Amsterdam
c.a.naesseth@uva.nl

## Abstract

Conventional diffusion models often rely on a fixed forward process, which implicitly defines complex marginal distributions over latent variables. This can often complicate the reverse process' task in learning generative trajectories, and results in costly inference for diffusion models. To address these limitations, we introduce Neural Flow Diffusion Models (NFDM), a novel framework that enhances diffusion models by supporting a broader range of forward processes beyond the standard linear Gaussian. We also propose a novel parameterization technique for learning the forward process. Our framework provides an end-to-end, simulation-free optimization objective, effectively minimizing a variational upper bound on the negative log-likelihood. Experimental results demonstrate NFDM's strong performance, evidenced by state-of-the-art likelihoods across a range of image generation tasks. Furthermore, we investigate NFDM's capacity for learning generative dynamics with specific characteristics, such as deterministic straight lines trajectories, and demonstrate how the framework can be adopted for learning bridges between two distributions. The results underscores NFDM's versatility and its potential for a wide range of applications.

## 1 Introduction

Diffusion models [56, 20] are a class of generative models constructed by two key processes: the forward process and the reverse process. The forward process gradually corrupts the data distribution, transforming it from its original form to a noised state. The reverse process learns to invert corruptions of the forward process and restore the data distribution. This way, the model learns to generate data from pure noise. Diffusion models have demonstrated remarkable results in various domains [21, 48, 63, 69, 65]. Nevertheless, most existing diffusion models fix the forward process to be predefined, usually linear, Gaussian which makes it unable to adapt to the task at hand or simplify the target for the reverse process. At the same time there is a growing body of work that demonstrates how modifications of the forward process improve performance in terms of generation quality [39, 66, 9], likelihood estimation [28, 40, 4] or sampling speed [32, 44, 64].

In this paper, we present Neural Flow Diffusion Models (NFDM), a framework that allows for the pre-specification and learning of complex latent variable distributions defined by the forward process. Unlike conventional diffusion models [20], which rely on a conditional Gaussian forward process, NFDM can accommodate any continuous (and learnable) distribution that can be expressed as an invertible mapping applied to noise. We also derive, and leverage, a new end-to-end simulation-free optimization procedure for NFDM, that minimizes a variational upper bound on the negative log-likelihood (NLL).

Furthermore, we propose an efficient neural network-based parameterization for the forward process, enabling it to adapt to the reverse process during training and simplify the learning of the data distribution. To demonstrate NFDM's capabilities with a learnable forward process we provide

38th Conference on Neural Information Processing Systems (NeurIPS 2024).

experimental results on CIFAR-10, ImageNet 32 and 64, attaining state-of-the-art NLL results, which is crucial for many applications such as data compression [22, 71], anomaly detection [7, 13] and out-of-distribution detection [51, 70].

Then, leveraging the flexibility of NFDM, we demonstrate how this framework can be applied to learn bridges between two distributions using the AFHQ dataset. Finally, we explore training with constraints on the reverse process to learn generative dynamics with specific properties. As a case study, we discuss curvature and obstacle avoidance penalties on the deterministic generative trajectories. Our empirical results indicate improved computational efficiency compared to baselines on CIFAR-10, downsampled ImageNet, and synthetic data.

We summarize our contributions as follows:

1. We introduce Neural Flow Diffusion Models (NFDM), improving diffusion modelling through a learnable forward process.

2. We develop an end-to-end optimization procedure that minimizes an upper bound on the negative log-likelihood in a simulation-free manner.

3. We demonstrate state-of-the-art log-likelihood results on CIFAR-10, ImageNet 32 and 64.

4. We show how NFDM can be used in learning bridges and generative processes with specific properties, such as dynamics with straight line trajectories, leading to significantly faster sampling speeds and enhanced generation quality with fewer sampling steps.

## 2  Background

Diffusion models are generative latent variable models consisting of two processes: the forward and the reverse (or generative) process. The forward process is a dynamic process that takes a data point $\mathbf{x} \sim q(\mathbf{x}), \mathbf{x} \in \mathbb{R}^D$, and perturbs it over time by injecting noise. This generates a trajectory of latent variables $\{\mathbf{z}(t)\}_{t \in [0,1]}$, conditional on the data $\mathbf{x}$, where $[0, 1]$ is a fixed time horizon and $\mathbf{z}_t = \mathbf{z}(t) \in \mathbb{R}^D$. The (conditional) distribution can be described by an initial distribution $q(\mathbf{z}_0|\mathbf{x})$ and a Stochastic Differential Equation (SDE) with a linear drift term $\tilde{f}^F(\mathbf{z}_t, t) : \mathbb{R}^D \times [0, 1] \mapsto \mathbb{R}^D$, scalar variance $g(t) : [0, 1] \mapsto \mathbb{R}_+$, and a standard Wiener process $\mathbf{w}$:

$$d\mathbf{z}_t = \tilde{f}^F(\mathbf{z}_t, t)dt + g(t)d\mathbf{w}. \tag{1}$$

Due to the linearity of $\tilde{f}^F$, we can reconstruct the conditional marginal distribution $q(\mathbf{z}_t|\mathbf{x}) = \mathcal{N}(\mathbf{z}_t; \alpha_t \mathbf{x}, \sigma_t^2 I)$. Typically, the conditional distributions evolve from some low variance distribution $q(\mathbf{z}_0|\mathbf{x}) \approx \delta(\mathbf{z}_0 - \mathbf{x})$ to a unit Gaussian $q(\mathbf{z}_1|\mathbf{x}) \approx \mathcal{N}(\mathbf{z}_1; 0, I)$. This forward process is then reversed by starting from the prior $\mathbf{z}_1 \sim \mathcal{N}(\mathbf{z}_1; 0, I)$, and following the reverse SDE [3]:

$$d\mathbf{z}_t = \tilde{f}^B(\mathbf{z}_t, t)dt + g(t)d\bar{\mathbf{w}}, \quad \text{where} \quad \tilde{f}^B(\mathbf{z}_t, t) = \tilde{f}^F(\mathbf{z}_t, t) - g^2(t)\nabla_{\mathbf{z}_t} \log q(\mathbf{z}_t). \tag{2}$$

Here, $\bar{\mathbf{w}}$ denotes a standard Wiener process where time flows backwards. Diffusion models approximate this reverse process by learning $\nabla_{\mathbf{z}_t} \log q(\mathbf{z}_t)$, known as the score function, through a $\lambda_t$-weighted denoising score matching loss:

$$\mathbb{E}_{u(t)q(\mathbf{x}, \mathbf{z}_t)} \left[ \lambda_t \big\| s_\theta(\mathbf{z}_t, t) - \nabla_{\mathbf{z}_t} \log q(\mathbf{z}_t|\mathbf{x}) \big\|_2^2 \right], \tag{3}$$

where $u(t)$ represents a uniform distribution over the interval $[0, 1]$, and $s_\theta : \mathbb{R}^D \times [0, 1] \mapsto \mathbb{R}^D$ is a learnable approximation. With a learned score function $s_\theta(\mathbf{z}_t, t)$, one can generate a sample from the reverse process by first sampling from the prior $\mathbf{z}_1 \sim \mathcal{N}(\mathbf{z}_1; 0, I)$, and then simulating the reverse SDE, resulting in a sample $\mathbf{z}_0 \sim p_\theta(\mathbf{z}_0) \approx q(\mathbf{z}_0) \approx q(\mathbf{x})$:

$$d\mathbf{z}_t = \big[ \tilde{f}^F(\mathbf{z}_t, t) - g^2(t)s_\theta(\mathbf{z}_t, t) \big] dt + g(t)d\bar{\mathbf{w}}. \tag{4}$$

Diffusion models possess several important properties. For example, for a specific $\lambda_t$, the objective (eq. (3)) can be reformulated [59] as an Evidence Lower Bound (ELBO) on the model's likelihood. Furthermore, the minimization of denoising score matching (eq. (3)) is a simulation-free procedure. This means that simulating either the forward or reverse processes through its SDE is not necessary for sampling $\mathbf{z}_t$, nor is it necessary for estimating the gradient of the loss function. Instead, we can

directly sample $\mathbf{z}_t \sim q(\mathbf{z}_t|\mathbf{x})$. The simulation-free nature of this approach is a crucial aspect for efficient optimization.

Another notable property is the existence of an Ordinary Differential Equation (ODE) corresponding to the same marginal densities $q(\mathbf{z}_t)$ as the SDE (eq. (1)):

$$d\mathbf{z}_t = f(\mathbf{z}_t, t)dt, \quad \text{where} \tag{5}$$

$$f(\mathbf{z}_t, t) = \tilde{f}^F(\mathbf{z}_t, t) - \frac{g^2(t)}{2}\nabla_{\mathbf{z}_t} \log q_\varphi(\mathbf{z}_t). \tag{6}$$

This implies that we can sample from diffusion models deterministically, allowing the use of off-the-shelf numerical ODE solvers for sampling, which may improve the sampling speed compared to stochastic sampling that requires simulating an SDE. Additionally, deterministic sampling enables us to compute densities by treating the model as a continuous normalizing flow, as detailed in [6, 17].

## 3 Neural Flow Diffusion Models

In diffusion models the forward process defines stochastic conditional trajectories $\{\mathbf{z}(t)\}_{t\in[0,1]}$ and the reverse process tries to match the marginal distribution of trajectories. This construction can be viewed as a specific type of hierarchical Variational Autoencoders (VAEs) [30, 45]. However, in conventional diffusion models the latent variables are inferred through a pre-specified linear combination of the data point and Gaussian noise. This formulation limits diffusion models in terms of the flexibility of their latent space, and makes learning of the reverse process more challenging. To address this limitation, we propose a generalized form of the forward process that enables the definition and learning of a broad range of distributions in the latent space. From a practical perspective, a more flexible forward process can simplify the task of learning the reverse process. From a theoretical perspective, learning of the forward process is analogous to learning the variational distribution in a hierarchical VAE, which gives a tighter bound on model's NLL (see an extended discussion in Appendix B.1).

In this section, we introduce Neural Flow Diffusion Models (NFDM) – a framework that generalizes conventional diffusion models. The key idea in NFDM is to define the forward process' conditional SDE implicitly via a learnable transformation $F_\varphi(\varepsilon, t, \mathbf{x})$ that defines the marginal distributions. This lets the user define a broad range of continuous time- and data-dependent forward processes, that the reverse process will learn to invert. Importantly, NFDM retains crucial properties of conventional diffusion models, like likelihood-based and simulation-free training. Previous diffusion models emerge as special cases when the data transformation is linear, time-independent, and/or additive Gaussian.

### 3.1 Forward Process

We approach the forward process constructively. The ultimate goal is a learnable distribution over trajectories, $\{\mathbf{z}(t)\}_{t\in[0,1]}$ given $\mathbf{x}$, realized by a conditional SDE constructed in three steps.

First, we (implicitly) define the conditional marginal distributions $q_\varphi(\mathbf{z}_t|\mathbf{x})$ for $t \in [0, 1]$ using $F_\varphi(\varepsilon, t, \mathbf{x})$. Then, we introduce the corresponding conditional ODE that together with an initial distribution $q_\varphi(\mathbf{z}_0|\mathbf{x})$ matches the conditional marginal distribution $q_\varphi(\mathbf{z}_t|\mathbf{x})$. Finally, we define a conditional SDE that defines a distribution over trajectories $\{\mathbf{z}(t)\}_{t\in[0,1]}$, with marginal distributions $q_\varphi(\mathbf{z}_t|\mathbf{x})$ for each $t$.

**Forward Marginal Distribution**. We characterize the marginal distribution $q_\varphi(\mathbf{z}_t|\mathbf{x})$ of the forward process trough a function that transforms a noise sample $\varepsilon$ into $\mathbf{z}_t$, conditional on the time step $t$ and data point $\mathbf{x}$:

$$\mathbf{z}_t = F_\varphi(\varepsilon, t, \mathbf{x}), \tag{7}$$

where $F_\varphi : \mathbb{R}^D \times [0, 1] \times \mathbb{R}^D \mapsto \mathbb{R}^D$ and $\varepsilon \sim q(\varepsilon) = \mathcal{N}(\varepsilon; 0, I)$. This defines the conditional distribution of latent variables $q_\varphi(\mathbf{z}_t|\mathbf{x})$. Additionally, eq. (7) facilitates direct and efficient sampling from $q_\varphi(\mathbf{z}_t|\mathbf{x})$ through $F_\varphi$ in eq. (7).

**Conditional ODE**. We assume that $F_\varphi$ is differentiable with respect to $\varepsilon$ and $t$, invertible with respect to $\varepsilon$. Further, we assume that fixing specific values of $\mathbf{x}$ and $\varepsilon$ and varying $t$ from 0 to 1 results in a

smooth trajectory from $\mathbf{z}_0$ to $\mathbf{z}_1$. Differentiating these trajectories over time yields a velocity field corresponding to the conditional distribution $q_\varphi(\mathbf{z}_t|\mathbf{x})$, thereby defining a conditional ODE:

$$d\mathbf{z}_t = f_\varphi(\mathbf{z}_t, t, \mathbf{x})dt, \quad \text{where} \quad f_\varphi(\mathbf{z}_t, t, \mathbf{x}) = \left.\frac{\partial F_\varphi(\varepsilon, t, \mathbf{x})}{\partial t}\right|_{\varepsilon = F_\varphi^{-1}(\mathbf{z}_t, t, \mathbf{x})}. \tag{8}$$

Therefore, if we sample $\mathbf{z}_0 \sim q_\varphi(\mathbf{z}_0|\mathbf{x})$ and solve eq. (8) until time $t$, we have $\mathbf{z}_t \sim q_\varphi(\mathbf{z}_t|\mathbf{x})$.

The time derivative of $F_\varphi$ may be calculated efficiently using automatic differentiation tools like PyTorch [42] or JAX [5] (see details in Appendix B.2).

**Conditional SDE**. The function $F_\varphi$ and the distribution of the noise $q(\varepsilon)$ together defines $q_\varphi(\mathbf{z}_t|\mathbf{x})$. To completely define the distribution of trajectories $\{\mathbf{z}(t)\}_{t\in[0,1]}$, we introduce a conditional SDE that starts from sample $\mathbf{z}_0$ and runs forward in time.

With access to both the ODE and score function $\nabla_{\mathbf{z}_t} \log q_\varphi(\mathbf{z}_t|\mathbf{x})$, an SDE [61] with marginal distributions $q_\varphi(\mathbf{z}_t|\mathbf{x})$, is:

$$d\mathbf{z}_t = \tilde{f}_\varphi^F(\mathbf{z}_t, t, \mathbf{x})dt + g_\varphi(t)d\mathbf{w}, \quad \text{where} \tag{9}$$

$$\tilde{f}_\varphi^F(\mathbf{z}_t, t, \mathbf{x}) = f_\varphi(\mathbf{z}_t, t, \mathbf{x}) + \frac{g_\varphi^2(t)}{2}\nabla_{\mathbf{z}_t} \log q_\varphi(\mathbf{z}_t|\mathbf{x}).$$

Here, $g_\varphi : [0, 1] \mapsto \mathbb{R}_+$ is a scalar function, and $\mathbf{w}$ represents a standard Wiener process. Note that $g_\varphi$ only influences the distribution of trajectories $\{\mathbf{z}(t)\}_{t\in[0,1]}$. The marginal distributions are the same, $q_\varphi(\mathbf{z}_t|\mathbf{x})$, for any choice of $g_\varphi$.

The SDE in eq. (9) requires access to the conditional score function $\nabla_{\mathbf{z}_t} \log q_\varphi(\mathbf{z}_t|\mathbf{x})$, which in the general case can be computationally costly. However, for $F_\varphi$ that allow efficient evaluation of the log-determinant of the Jacobian matrix, the score function can be calculated efficiently. Examples of such $F_\varphi$ are functions linear with respect to $\varepsilon$ or RealNVP style architectures [14, 29]. The calculation of this score function is further discussed in Appendix B.3.

## 3.2 Reverse (Generative) Process

To define the reverse (generative) process, we specify a reverse SDE that starts from $\mathbf{z}_1 \sim p(\mathbf{z}_1)$ and runs backwards in time. To do so we first introduce a conditional reverse SDE that reverses the conditional forward SDE (eq. (9)). Following [3], we define:

$$d\mathbf{z}_t = \tilde{f}_\varphi^B(\mathbf{z}_t, t, \mathbf{x})dt + g_\varphi(t)d\bar{\mathbf{w}}, \quad \text{where} \tag{10}$$

$$\tilde{f}_\varphi^B(\mathbf{z}_t, t, \mathbf{x}) = f_\varphi(\mathbf{z}_t, t, \mathbf{x}) - \frac{g_\varphi^2(t)}{2}\nabla_{\mathbf{z}_t} \log q_\varphi(\mathbf{z}_t|\mathbf{x}).$$

Secondly, leveraging this reverse SDE (see eq. (10)) we incorporate a prediction of $\mathbf{x}$:

$$d\mathbf{z}_t = \hat{f}_{\theta,\varphi}(\mathbf{z}_t, t)dt + g_\varphi(t)d\bar{\mathbf{w}}, \quad \text{where} \quad \hat{f}_{\theta,\varphi}(\mathbf{z}_t, t) = \tilde{f}_\varphi^B(\mathbf{z}_t, t, \hat{\mathbf{x}}_\theta(\mathbf{z}_t, t)), \tag{11}$$

and $\hat{\mathbf{x}}_\theta : \mathbb{R}^D \times [0, 1] \mapsto \mathbb{R}^D$ is a function that predicts the data point $\mathbf{x}$. This SDE defines the dynamics of the generative trajectories $\{\mathbf{z}(t)\}_{t\in[0,1]}$.

To fully specify the reverse process, it is also necessary to define a prior distribution $p(\mathbf{z}_1)$ and a reconstruction distribution $p(\mathbf{x}|\mathbf{z}_0)$. In all of our experiments, we set the prior $p(\mathbf{z}_1)$ to be a unit Gaussian distribution $\mathcal{N}(\mathbf{z}_1; 0, I)$ and let $p(\mathbf{x}|\mathbf{z}_0)$ be a Gaussian distribution with a small variance $\mathcal{N}(\mathbf{x}; \mathbf{z}_0, \delta^2 I)$, where $\delta^2 = 10^{-4}$.

The above parameterization of the reverse process is not the only possibility. However, it is a convenient choice as it allows for the definition of the reverse process simply through the prediction of $\mathbf{x}$, similar to conventional diffusion models [20]. We leave exploration of alternate parameterizations for future work.

| **Algorithm 1** Optimization of NFDM | **Algorithm 2** Stochastic Sampling from NFDM |
|---|---|
| **Require:** $q(\mathbf{x})$, $F_\varphi$, $g_\varphi$, $\hat{\mathbf{x}}_\theta$ | **Require:** $F_\varphi$, $g_\varphi$, $\hat{\mathbf{x}}_\theta$, $T$ – number of steps |
|   **for** learning iterations **do** |   $\Delta t = \frac{1}{T}$, $\mathbf{z}_1 \sim p(\mathbf{z}_1)$ |
|     $\mathbf{x} \sim q(\mathbf{x})$, $t \sim u(t)$ |     **for** $t = 1, \ldots, \frac{2}{T}, \frac{1}{T}$ **do** |
|     $\mathbf{z}_t \sim q_\varphi(\mathbf{z}_t\|\mathbf{x})$ |       $\bar{\mathbf{w}} \sim \mathcal{N}(0, I)$ |
|     $\mathcal{L} = \frac{1}{2g_\varphi^2(t)}\big\|\tilde{f}_\varphi^B(\mathbf{z}_t, t, \mathbf{x}) - \hat{f}_{\theta,\varphi}(\mathbf{z}_t, t)\big\|_2^2$ |       $\mathbf{z}_{t-\Delta t} = \mathbf{z}_t - \hat{f}_{\theta,\varphi}(\mathbf{z}_t, t)\Delta t + g_\varphi(t)\bar{\mathbf{w}}\sqrt{\Delta t}$ |
|     Gradient step on $\theta$ and $\varphi$ w.r.t. $\mathcal{L}$ |   **end for** |
|   **end for** |   $\mathbf{x} \sim p(\mathbf{x}\|\mathbf{z}_0)$ |

## 3.3 Optimization and Sampling

With $F_\varphi$ in eq. (7) parameterized such that $q_\varphi(\mathbf{z}_0|\mathbf{x}) \approx \delta(\mathbf{x} - \mathbf{z}_0)$ and $q_\varphi(\mathbf{z}_1|\mathbf{x}) \approx p(\mathbf{z}_1)$, we propose to optimize the forward and reverse processes of NFDM jointly, minimizing the following objective:

$$\mathcal{L} = \mathbb{E}_{u(t)q(\mathbf{x})q_\varphi(\mathbf{z}_t|\mathbf{x})}\left[\frac{1}{2g_\varphi^2(t)}\big\|\tilde{f}_\varphi^B(\mathbf{z}_t, t, \mathbf{x}) - \hat{f}_{\theta,\varphi}(\mathbf{z}_t, t)\big\|_2^2\right]. \tag{12}$$

Since the forward process is parameterized by $\varphi$, we need to optimize the objective (eq. (12)) with respect to both $\varphi$ and $\theta$ jointly. We discuss parameterization of the forward process in Appendix B.5.

As we demonstrate in Appendix A.1 the objective $\mathcal{L}$ shares similarities with the standard diffusion model objective [20], and it provides a variational bound on the model's log-likelihood $\log p_{\theta,\varphi}(\mathbf{x})$. The objective $\mathcal{L}$ also exhibits strong connections with both the Flow Matching [33] and Score Matching [68] objectives. We explore these connections in Appendix E.4, where we also discuss the role of $g_\varphi$ in detail. It is important to note that despite the parameterization of the reverse process through the prediction of $\mathbf{x}$, the objective $\mathcal{L}$ optimizes the generative dynamics and does not necessarily lead to accurate predictions of $\mathbf{x}$.

A key characteristic of the NFDM objective is its compatibility with the simulation-free paradigm, which is critical for efficient optimization. We summarize the training procedure in Algorithm 1.

To sample from the trained reverse process we can simulate the SDE, as defined in Section 3.2. This procedure is summarized in Algorithm 2. Additionally, during the sampling process, we can adjust the level of stochasticity by modifying $g_\varphi(t)$ (eq. (9)). It is important to note that changes to $g_\varphi(t)$ also influence $\tilde{f}_\varphi^F$ (eq. (9)) and $\hat{f}_{\theta,\varphi}$ (eq. (11)). In the extreme case where $g_\varphi(t) \equiv 0$, the reverse process becomes deterministic, allowing us to utilize off-the-shelf numerical ODE solvers and to estimate densities [6, 17]. We provide an extended sampling discussion in Appendix B.4.

## 4 Neural Flow Bridge Models

In this section we discuss a simple modification to the NFDM framework that enables us to learn bridges between two data distributions, $q(\mathbf{x}_0)$ and $q(\mathbf{x}_1)$, a modification we refer to as Neural Flow Bridge Models (NFBM).

In the context of bridges, we consider a joint data distribution $q(\mathbf{x}_0, \mathbf{x}_1)$ (which may be factorized as $q(\mathbf{x}_0)q(\mathbf{x}_1)$ if paired data is unavailable). Our goal is to learn a generative process that starts from $\mathbf{x}_1$ and generates $\mathbf{x}_0$ such that $q(\mathbf{x}_0) \approx p_\theta(\mathbf{x}_0) = \int q(\mathbf{x}_1)p_\theta(\mathbf{x}_0|\mathbf{x}_1)d\mathbf{x}_1$. To turn NFDM into NFBM, we modify both the forward and reverse processes. In NFDM, the forward process is defined by two functions: $F_\varphi(\varepsilon, t, \mathbf{x})$ (see eq. (7)) and $g_t$ (see eq. (9)). For NFBM, we let $F_\varphi$ depend on both $\mathbf{x}_0$ and $\mathbf{x}_1$, thereby conditioning the entire forward process on these data points:

$$\mathbf{z}_t = F_\varphi(\varepsilon, t, \mathbf{x}_0, \mathbf{x}_1), \tag{13}$$

Similar to the discussion in Section 3.2, for the reverse process of NFBM, we predict the data points $\mathbf{x}_0$ and $\mathbf{x}_1$ from $\mathbf{z}_t$ and $t$, and substitute them into the conditional reverse time SDE (eq. (10)):

$$d\mathbf{z}_t = \hat{f}_{\theta,\varphi}(\mathbf{z}_t, t)dt + g_\varphi(t)d\bar{\mathbf{w}}, \quad \text{where} \quad \hat{f}_{\theta,\varphi}(\mathbf{z}_t, t) = \tilde{f}_\varphi^B\big(\mathbf{z}_t, t, \hat{\mathbf{x}}_\theta^{0,1}(\mathbf{z}_t, t)\big). \tag{14}$$

Here, the function $\hat{\mathbf{x}}_\theta^{0,1}(\mathbf{z}_t, t)$ returns predictions of both $\mathbf{x}_0$ and $\mathbf{x}_1$. Alternatively, the data point $\mathbf{x}_1$ can be reused as conditioning for intermediate steps of the reverse process instead of predicting both points, a strategy further detailed in Appendix C.2.

When $F_\varphi$ in eq. (13) is parameterized such that $q_\varphi(\mathbf{z}_0|\mathbf{x}_0, \mathbf{x}_1) \approx \delta(\mathbf{x} - \mathbf{z}_0)$ and $q_\varphi(\mathbf{z}_1|\mathbf{x}_0, \mathbf{x}_1) \approx p(\mathbf{z}_1|\mathbf{x}_1)$ (see Appendix C.1 for parameterization details of the NFBM forward process), we propose training NFBM by minimizing the following objective:

$$\mathcal{L} = \mathbb{E}_{u(t)q(\mathbf{x}_0,\mathbf{x}_1)q_\varphi(\mathbf{z}_t|\mathbf{x}_0,\mathbf{x}_1)} \left[ \frac{1}{2g_\varphi^2(t)} \left\| \tilde{f}_\varphi^B(\mathbf{z}_t, t, \mathbf{x}_0, \mathbf{x}_1) - \hat{f}_{\theta,\varphi}(\mathbf{z}_t, t) \right\|_2^2 \right]. \tag{15}$$

The derivation of this objective is provided in Appendix A.2. This objective shares key properties with the NFDM objective, it provides a variational bound on the model's log-likelihood $\log p_{\theta,\varphi}(\mathbf{x}_0)$, and is compatible with the simulation-free paradigm. Furthermore, it allows for sampling with various levels of stochasticity (by adjusting $g_\varphi(t)$) including deterministic sampling (when $g_\varphi(t) \equiv 0$).

Thus, the NFDM framework not only enables the construction of generative models, but also facilitates learning bridge models between two data distributions.

# 5   Restricted NFDM

We introduce NFDM as a powerful framework that enables learning of the forward process. However, there is in general an infinite number of forward and reverse processes that correspond to each other. In this section we discuss how the flexibility of NFDM (and NFBM) allows learning generative dynamic with user-specified beneficial properties.

Suppose our objective is to learn straight generative ODE trajectories, which can be highly beneficial, as it enables generation with far fewer steps in the ODE solver. One approach is to introduce penalties on the curvature of the reverse process's trajectories and let the learnable forward process to adapt to align with the reverse process. As a result, we would obtain a generative process that not only corresponds to the forward process, ensuring accurate data generation, but also features the desired property of straight trajectories. We discuss NFDM with restrictions in more details in Appendix D.1.

We propose learning a model with curvature penalty as suggested by [25]:

$$\mathcal{L}_{\text{OT}} = \mathcal{L} + \lambda \mathcal{L}_{\text{crv}}, \quad \text{where} \quad \mathcal{L}_{\text{crv}} = \mathbb{E}_{u(t)q_\varphi(\mathbf{x},\mathbf{z}_t)} \left\| \frac{d\hat{f}_{\theta,\varphi}(\mathbf{z}_t, t)}{dt} \right\|_2^2. \tag{16}$$

We refer to this variant as NFDM-OT[1]. $\mathcal{L}_{\text{crv}}$ is an additional curvature loss that penalizes the second time derivative of the generative ODE trajectories. $\mathcal{L}$ is estimated as in Section 3.3, whereas when calculating $\mathcal{L}_{\text{crv}}$ we set $g_\varphi(t) \equiv 0$. $\mathcal{L}_{\text{crv}} \equiv 0$ ensures that the generative trajectories are straight lines. In our experiments, we set $\lambda = 10^{-2}$. Empirical evidence suggests that higher values of $\lambda$ lead to slower convergence of the model. The specifics of the curvature loss are elaborated upon in more detail in Appendix D.2.

We would like to emphasise that the purpose of this section is to provide an example of how the NFDM framework allows learning dynamics with specific properties. Not to propose the best way for learning generative straight-line generative dynamics. In addition, we note that conventional diffusion models are incapable of handling such penalization strategies. In diffusion models with a fixed forward process, the target for the reverse process is predetermined and the corresponding ODE trajectories are highly curved. Hence, imposing constraints on the reverse process, such as trajectory straightness, would lead to a mismatch with the forward process and, consequently, an inability to generate samples with high data fidelity.

# 6   Experiments

We first showcase results demonstrating that NFDM consistently achieves better likelihood compared to baselines, obtaining state-of-the-art diffusion modeling results on the CIFAR-10 [31] and down-sampled ImageNet [11, 67] datasets. Then, we explore the NFDM-OT modification, which penalizes

---

[1]OT stands for optimal transport. This designation is used for convenience. While straight trajectories are a necessary condition for dynamic optimal transport, they are not sufficient.

Table 1: Comparison of NFDM results with baselines on density estimation tasks. We present results in terms of BPD, lower is better. NFDM achieves state-of-the-art results across all three benchmark tasks.

| Model | CIFAR10 | ImageNet 32 | ImageNet 64 |
|---|---|---|---|
| DDPM [20] | 3.69 | | |
| Score SDE [61] | 2.99 | | |
| Improved DDPM [39] | 2.94 | | 3.54 |
| VDM [28] | 2.65 | 3.72 | 3.40 |
| Score Flow [59] | 2.83 | 3.76 | |
| Flow Matching [33] | 2.99 | 3.53 | 3.31 |
| Stochastic Interp. [2] | 2.99 | 3.48 | |
| i-DODE [76] | 2.56 | 3.43 | |
| NDM [4] | 2.70 | 3.55 | 3.35 |
| MuLAN [49] | 2.55 | 3.67 | |
| NFDM (**Gaussian** $q_\varphi(\mathbf{z}_t|\mathbf{x})$) | 2.49 | 3.36 | **3.20** |
| NFDM (**non-Gaussian** $q_\varphi(\mathbf{z}_t|\mathbf{x})$) | **2.48** | **3.34** | **3.20** |

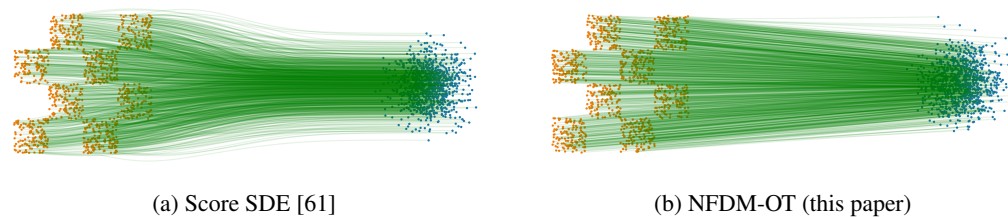

(a) Score SDE [61]  (b) NFDM-OT (this paper)

Figure 1: Comparison of trajectories between the data distribution (on the left) and the prior distribution (on the right), as learned by conventional diffusion and NFDM-OT.

the curvature of the deterministic generative trajectories. The NFDM-OT reduces trajectory curvature, significantly lowering the number of generative steps required for sampling. Finally, we demonstrate the NFBM modification, learning bridges on the AFHQ [8] dataset and several synthetic examples.

We report the NLL in Bits Per Dimension (BPD) and sample quality measured by the Frechet Inception Distance (FID) score [19]. The NLL is calculated by integrating the ODEs using the RK45 solver [15], with all NLL metrics computed on test data. For FID, we provide the average over 50k generated images.

Unless otherwise stated, we parameterized the forward process of NFDM and NFBM such that $q_\varphi(\mathbf{z}_t|\mathbf{x})$ is a Gaussian with learnable mean and covariance (see details in Appendices B.5 and C.1). For a detailed description of parameterizations and other experimental details, please refer to Appendix F.

The primary aim of the experiments with NFDM-OT and NFBM is not to introduce a novel model that surpasses others in few-step generation or learning bridges, but rather to showcase the ability of the NFDM framework to learn generative dynamics with specific properties. The straightness of the trajectories is just one example of such a property. We leave it for future research to explore different parameterizations and modified objectives for NFDM that may yield even better results.

Please refer to appendix G.3 for generated samples. The code is available at `https://github.com/GrigoryBartosh/neural_diffusion`.

## 6.1 Likelihood Estimation

For the first experiments, in addition to Gaussian we also provide results with non-Gaussian parameterization of the forward process (see Appendix B.5). Table 1 summarizes the NLL results on the CIFAR-10 and two downsampled ImageNet datasets. Notably, NFDM outperforms diffusion-based baselines on all three datasets with both parameterizations, achieving state-of-the-art performance. The improved performance due to NFDM is a natural progression for the following reasons.

Table 2: Summary of FID results for few-step generation. The table is divided into three sections, based on different types of methods: those that do not minimize curvature, solvers for pretrained models, and models that specifically aim to minimize curvature. For the DDPM, we include results corresponding to two distinct objectives: the full ELBO-based objective and a simplified objective ($\mathcal{L}_{\text{simple}}$). NFDM-OT outperforms baselines with comparable NFE values.

| Model | **CIFAR-10** | | **ImageNet 32** | | **ImageNet 64** | |
|---|---|---|---|---|---|---|
| | NFE ↓ | FID ↓ | NFE ↓ | FID ↓ | NFE ↓ | FID ↓ |
| DDPM ($\mathcal{L}_{\text{simple}}$) [20] | 1000 | 3.17 | | | | |
| DDPM (ELBO) [20] | 1000 | 13.51 | | | | |
| Flow Matching [33] | 142 | 6.35 | 122 | 5.02 | 138 | 14.14 |
| DDIM [57] | 10 | 13.36 | | | | |
| DPM Solver [38] | 12 | 5.28 | | | | |
| | 24 | 2.75 | | | | |
| Trajectory Curvature Minimization [32] | 5 | 18.74 | | | | |
| Multisample Flow Matching [44] | | | 4 | 17.28 | 4 | 38.45 |
| | | | 12 | 7.18 | 12 | 17.6 |
| NFDM-OT (**this paper**) | 2 | 12.44 | 2 | 9.83 | 2 | 27.70 |
| | 4 | 7.76 | 4 | 6.13 | 4 | 17.28 |
| | 12 | 5.20 | 12 | 4.11 | 12 | 11.58 |

First, it is well-established [59, 76] that diffusion models exhibit improved likelihood estimation when trained with the full ELBO objective. The objective $\mathcal{L}$ (eq. (12)) used for training the NFDM is also a variational bound on the likelihood.

Second, diffusion models can be seen as hierarchical VAEs. From this perspective, most baselines in Table 1 resemble VAEs with either fixed or constrained variational distributions. In contrast, the NFDM extends beyond these baselines by providing a more flexible variational distribution. That is true even for the Gaussian parameterization with a mean and covariance that is non-linear in $\mathbf{x}$ and $t$. This flexibility allows the NFDM to better conform to the reverse process, consequently enhancing likelihood estimation.

## 6.2 Straight Trajectories

We next evaluate NFDM-OT, designed to penalize the deterministic generative trajectory curvature. First, we compare NFDM-OT with a conventional continuous-time diffusion model. Figure 1a illustrates deterministic trajectories between a two-dimensional data distribution and a unit Gaussian distribution learnt by a conventional diffusion model [61]. Figure 1b depicts trajectories learnt by NFDM-OT. Conventional diffusion, being constrained in its forward process, learns highly curved trajectories, whereas NFDM-OT successfully learns straight generative trajectories as desired. In Appendix G.1 we provide some additional reult, demonstrating the importance of learnable forward process for NFDM-OT.

Then, we present results of NFDM-OT on image datasets. Table 2 reports the FID scores for 2, 4, and 12 Number of Function Evaluations (NFE) with respect to the function $\hat{f}_{\theta,\varphi}$ (eq. (11)). In this experiment, we employ Euler's method for sampling integration. For the specified NFEs, NFDM-OT demonstrates superior sample quality compared to other approaches with similar NFE values. Specifically, NFDM-OT outperforms approaches specifically designed to minimize the curvature of generative trajectories [44, 32].

Importantly, NFDM-OT is trained with an ELBO-based objective (eq. (12)), which is known to yield higher FID scores for diffusion models [59, 76]. In contrast, some of the approaches listed in Table 2 are trained with different objectives, leading to improved FID scores. Even so, for comparable NFE values NFDM-OT still achieves superior results. We provide additional results of NFDM-OT in Appendix G.2.

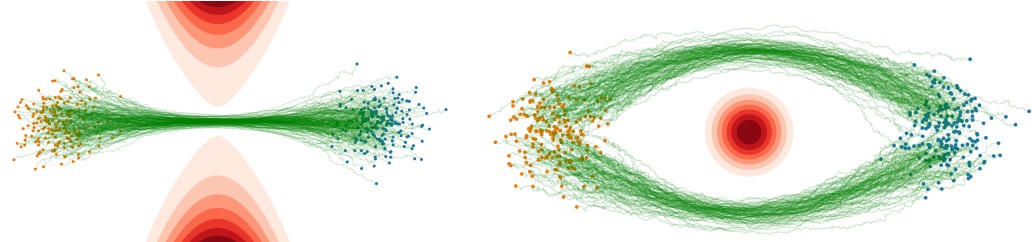

Figure 2: Trajectories of NFBM between two-dimensional data distributions, trained with an additional penalty to avoid obstacles.

## 6.3 Bridges

We consider the task of translating images of dogs into images of cats from the downsampled AFHQ dataset. Table 3 reports the FID scores on images sampled with 1000 steps. Notably, NFBM outperforms the baselines, demonstrating the effectiveness of our framework for learning bridges.

Finally, we demonstrate that the framework flexibility also allows for learning dynamics with specific properties in the case of NFBM. To illustrate this, we adopted the strategy discussed in Section 5 and learn NFBM with an additional penalty to avoid obstacles (see

Table 3: FID values of dog $\rightarrow$ cat on AFHQ $64$.

| Model | FID $\downarrow$ |
|---|---|
| DSBM [54] | 14.16 |
| GSBM [34] | 12.39 |
| NFBM | **12**.**06** |

details in Appendix F.2). Figure 2 visualizes the learned stochastic trajectories. It is evident that NFBM has efficiently learned to translate distributions while avoiding obstacles.

## 7 Related Work

Diffusion models, originally proposed by [56], have evolved significantly through subsequent developments [60, 20]. These advancements have resulted in remarkable generative quality for high-dimensional data distributions [12, 48]. Nevertheless, conventional diffusion models, typically relying on a linear Gaussian forward process, may not optimally fit some data distributions. To address this, alternative forward processes have been explored, such as alternative linear transformations [16, 55], combining blurring with Gaussian noise injection [46, 9, 23], diffusion in the wavelet spectrum [43], and forward processes based on the exponential family [41]. These models are limited by their fixed forward processes and may be seen as specific instances of NFDM.

Some studies have focused on making the forward process learnable. Approaches include a learnable noise injection schedule [28, 39, 49] and learning data transformations like time-independent transformations based on VAEs [66, 47] and normalizing flows [26], or time-dependent transformations [18, 55, 40, 4]. These methods can also be considered special cases of NFDM with specific transformations $F_\varphi$ (eq. (7)).

To improve the sampling efficiency, in orthogonal line of works alternative sampling methods have been studied [62, 35, 38, 53]. Additionally, distillation techniques have been applied [50, 58, 75, 72] to enhance sampling speed, albeit at the cost of training additional models. Most of these approches are compatible with NFDM and may be combined for further gains.

Another line of works proposed technics for learning straighter generative trajectories. However, these approaches requires distillation [37, 36], solving unstable min-max problem [1], estimating an optimal coupling over minibatches [44, 64] of the entire dataset, which, for large datasets, may become uninformative, or can be viewed as specific instances [32, 52] of NFDM.

The connections between NFDM and some related works is discussed further in Appendix E.

## 8 Conclusion and Limitations

In this paper we introduced NFDM, a novel simulation-free framework for improved diffusion modeling through a learnable forward processes. NFDM outperforms baseline diffusion models on

standard benchmarks, showcasing its effectiveness. However, NFDM is not just another model, but rather it is a versatile framework that facilitates the predefined specification and learning of a forward process. Similarly, the penalty on the curvature of deterministic trajectories discussed in Section 5 represents just one specific example of the restrictions that NFDM can accommodate.

Nonetheless, the advantages of NFDM come with certain trade-offs. Once the forward process is parameterized using a neural network, this leads to approximately 2.2 times longer optimization iteration of NFDM compared to conventional diffusion models. Additionally, as discussed in Section 3.1, NFDM is limited to forward process parameterisations with functions that match specific conditions.

Despite these challenges, we are optimistic about the potential of NFDM. NFDM opens up significant prospects for exploring new generative dynamics. We believe that with alternative parameterizations, modifications of the objective, and the integration of orthogonal approaches, NFDM has the potential to achieve even better results across a range of data modalities.

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

# A Derivations

## A.1 Derivation of NFDM's Objective

Our goal is to derive a variational bound on the negative log-likelihood of NFDM. For the convenience of the following derivations we will consider the forward process as a conditional reverse SDE (eq. (10)) that starts from $\mathbf{z}_1 \sim q_\varphi(\mathbf{z}_1|\mathbf{x})$ and flows backwards in time:

$$d\mathbf{z}_t = \tilde{f}_\varphi^B(\mathbf{z}_t, t, \mathbf{x})dt + g_\varphi(t)d\bar{\mathbf{w}}. \tag{17}$$

Similarly to a conditional SDE (eq. (9)) flowing forwards in time, this SDE also corresponds to conditional marginal distribution $q_\varphi(\mathbf{z}_t|\mathbf{x})$, which is implicitly defined by the invertible transformation $\mathbf{z}_t = F_\varphi(\varepsilon, t, \mathbf{x})$ (eq. (7)), where $\varepsilon \sim q(\varepsilon)$.

For the reverse process we use the definition from Section 3.2 as reverse SDE (eq. (11)) that starts from the prior distribution $p(\mathbf{z}_1)$, flows backwards in time, and finally samples $\mathbf{x} \sim p(\mathbf{x}|\mathbf{z}_0)$:

$$d\mathbf{z}_t = \hat{f}_{\theta,\varphi}(\mathbf{z}_t, t)dt + g_\varphi(t)d\bar{\mathbf{w}}. \tag{18}$$

To derive the NFDM's variational bound on the NLL we discretize processes, derive an objective in discrete case and then consider the limit of discretization to switch back to continuous time.

We discretize the conditional reverse SDE in eq. (17) and the reverse SDE in eq. (18), transitioning from continuous-time trajectories $\{\mathbf{z}(t)\}_{t\in[0,1]}$ to discrete-time trajectories $\bar{\mathbf{z}}_0, \bar{\mathbf{z}}_{\frac{1}{T}}, \ldots, \bar{\mathbf{z}}_1$, where $T$ represents the number of discrete steps of size $\Delta t = \frac{1}{T}$.

The discrete conditional reverse process starts from distribution $\bar{q}_\varphi(\bar{\mathbf{z}}_1|\mathbf{x}) = q_\varphi(\bar{\mathbf{z}}_1|\mathbf{x})$ and follows conditional reverse Markov chain:

$$\bar{q}_\varphi(\bar{\mathbf{z}}_{t-\Delta t}|\bar{\mathbf{z}}_t, \mathbf{x}) = \mathcal{N}\Big(\bar{\mathbf{z}}_{t-\Delta t}; \bar{\mathbf{z}}_t - \Delta t\tilde{f}_\varphi^B(\bar{\mathbf{z}}_t, t, \mathbf{x}), \Delta t g_\varphi^2(t)I\Big). \tag{19}$$

Similarly, the discrete generative process shares distributions $\bar{p}(\bar{\mathbf{z}}_1) = p(\bar{\mathbf{z}}_1)$ and $\bar{p}(\mathbf{x}|\bar{\mathbf{z}}_0) = p(\mathbf{x}|\bar{\mathbf{z}}_0)$ with the continuous one. However, instead of following the reverse SDE (eq. (18)) it follows the reverse Markov chain:

$$\bar{p}_{\theta,\varphi}(\bar{\mathbf{z}}_{t-\Delta t}|\bar{\mathbf{z}}_t) = \mathcal{N}\Big(\bar{\mathbf{z}}_{t-\Delta t}; \bar{\mathbf{z}}_t - \Delta t\hat{f}_{\theta,\varphi}(\bar{\mathbf{z}}_t, t), \Delta t g_\varphi^2(t)I\Big). \tag{20}$$

It's important to note that once we discretize the trajectories, $\bar{q}_\varphi(\bar{\mathbf{z}}_t|\mathbf{x}) \neq q_\varphi(\bar{\mathbf{z}}_t|\mathbf{x})$ and $\bar{p}_{\theta,\varphi}(\bar{\mathbf{z}}_t) \neq p_{\theta,\varphi}(\bar{\mathbf{z}}_t)$ except for $t = 1$. However, this discretization corresponds to an Euler-Maruyama method of integrating SDEs. Thus, we know that as $\Delta t \to 0$, the discretized trajectories converge to the continuous ones.

Now, considering these discrete processes and recognizing that they are Markovian conditional and unconditional processes moving backwards in time, we can leverage a key result from [56, 20]:

$$\mathbb{E}_{q(\mathbf{x})}\left[-\log \bar{p}_{\theta,\varphi}(\mathbf{x})\right] \leq \bar{\mathcal{L}}_{\text{rec}} + \bar{\mathcal{L}}_{\text{diff}} + \bar{\mathcal{L}}_{\text{prior}}, \quad \text{where} \tag{21}$$

$$\bar{\mathcal{L}}_{\text{rec}} = \mathbb{E}_{\bar{q}_\varphi(\mathbf{x},\bar{\mathbf{z}}_0)}\left[-\log p(\mathbf{x}|\bar{\mathbf{z}}_0)\right], \tag{22}$$

$$\bar{\mathcal{L}}_{\text{diff}} = \sum_{t=\frac{1}{T}}^{T} \mathbb{E}_{\bar{q}_\varphi(\mathbf{x},\bar{\mathbf{z}}_t)}\left[D_{\text{KL}}\big(\bar{q}_\varphi(\bar{\mathbf{z}}_{t-\Delta t}|\bar{\mathbf{z}}_t, \mathbf{x})\|\bar{p}_{\theta,\varphi}(\bar{\mathbf{z}}_{t-\Delta t}|\bar{\mathbf{z}}_t)\big)\right], \tag{23}$$

$$\bar{\mathcal{L}}_{\text{prior}} = \mathbb{E}_{q(\mathbf{x})}\left[D_{\text{KL}}\big(\bar{q}_\varphi(\bar{\mathbf{z}}_1|\mathbf{x})\|\bar{p}(\bar{\mathbf{z}}_1)\big)\right]. \tag{24}$$

Equation (21) establishes a variational upper bound on the negative log-likelihood of discretized reverse process. Taking the limits of this upper bound, the reconstruction term $\mathcal{L}_{\text{rec}} = \bar{\mathcal{L}}_{\text{rec}}$ and the

prior term $\mathcal{L}_{\text{prior}} = \bar{\mathcal{L}}_{\text{prior}}$ retain their forms. However, we can rewrite the diffusion term:

$$\mathcal{L}_{\text{diff}} = \lim_{\Delta t \to 0} \bar{\mathcal{L}}_{\text{diff}} \tag{25}$$

$$= \lim_{\Delta t \to 0} \sum_{t=\frac{1}{T}}^{T} \mathbb{E}_{\bar{q}_\varphi(\mathbf{x},\bar{\mathbf{z}}_t)} \left[ D_{\text{KL}}\big(\bar{q}_\varphi(\bar{\mathbf{z}}_{t-\Delta t}|\bar{\mathbf{z}}_t,\mathbf{x}) \| \bar{p}_{\theta,\varphi}(\bar{\mathbf{z}}_{t-\Delta t}|\bar{\mathbf{z}}_t)\big) \right] \tag{26}$$

$$= \lim_{\Delta t \to 0} \sum_{t=\frac{1}{T}}^{T} \mathbb{E}_{\bar{q}_\varphi(\mathbf{x},\bar{\mathbf{z}}_t)} \left[ \frac{1}{2\Delta t g_\varphi^2(t)} \left\| \bar{\mathbf{z}}_t - \Delta t \hat{f}_{\theta,\varphi}(\bar{\mathbf{z}}_t,t) - \bar{\mathbf{z}}_t + \Delta t \tilde{f}_\varphi^B(\bar{\mathbf{z}}_t,t,\mathbf{x}) \right\|_2^2 \right] \tag{27}$$

$$= \lim_{\Delta t \to 0} \sum_{t=\frac{1}{T}}^{T} \mathbb{E}_{\bar{q}_\varphi(\mathbf{x},\bar{\mathbf{z}}_t)} \left[ \frac{\Delta t^2}{2\Delta t g_\varphi^2(t)} \left\| \tilde{f}_\varphi^B(\bar{\mathbf{z}}_t,t,\mathbf{x}) - \hat{f}_{\theta,\varphi}(\bar{\mathbf{z}}_t,t) \right\|_2^2 \right] \tag{28}$$

$$= \mathbb{E}_{u(t)\bar{q}_\varphi(\mathbf{x},\bar{\mathbf{z}}_t)} \left[ \frac{1}{2 g_\varphi^2(t)} \left\| \tilde{f}_\varphi^B(\bar{\mathbf{z}}_t,t,\mathbf{x}) - \hat{f}_{\theta,\varphi}(\bar{\mathbf{z}}_t,t) \right\|_2^2 \right], \tag{29}$$

where $u(t)$ is a uniform distribution over a unit interval.

Therefore, as the discretized trajectory tends to the continuous one when $\Delta t \to 0$, we can use the obtained limits as a variational bound on the negative log-likelihood of the continuous model:

$$\mathbb{E}_{q(\mathbf{x})}\left[ -\log p_{\theta,\varphi}(\mathbf{x}) \right] \leq \mathcal{L} = \mathcal{L}_{\text{rec}} + \mathcal{L}_{\text{diff}} + \mathcal{L}_{\text{prior}}. \tag{30}$$

In general case, when function $F_\varphi$ is actually parameterized by $\varphi$, the objective $\mathcal{L}$ implies that we must optimise all three terms with respect to parameters $\varphi$ and $\theta$. However, when $F_\varphi$ is parameterized such that $q_\varphi(\mathbf{z}_0|\mathbf{x}) \approx \delta(\mathbf{x} - \mathbf{z}_0)$ and $q_\varphi(\mathbf{z}_1|\mathbf{x}) \approx p(\mathbf{z}_1)$ for any $\varphi$, the reconstruction $\mathcal{L}_{\text{rec}}$ and prior $\mathcal{L}_{\text{prior}}$ terms are constants with respect to the parameters.

## A.2 Derivation of NFBM's Objective

Similarly to the derivation of the NFDM objective in Appendix A.1, here we also apply discretization to derive the objective of NFBM.

The conditional reverse process (eq. (17)) and it's discretization (eq. (19)) stays the same, except now the conditional reverse process are conditioned on a pair of points $\mathbf{x} = (\mathbf{x}_0, \mathbf{x}_1)$, sampled from a joined data distribution $(\mathbf{x}_0, \mathbf{x}_1) \sim q(\mathbf{x}_0, \mathbf{x}_1)$. The reverse process (eq. (18)) and its discretization (eq. (20)) also have the same structure. However, for NFBM the reverse process starts from distribution $\bar{p}(\mathbf{z}_1|\mathbf{x}_1) = p(\mathbf{z}_1|\mathbf{x}_1)$ and at the end samples from $\bar{p}(\mathbf{x}_0|\mathbf{z}_0) = p(\mathbf{x}_0|\mathbf{z}_0)$.

We derive the variational bound on NLL of NFBM:

$$\mathbb{E}_{q(\mathbf{x}_0)}\left[ -\log \bar{p}_{\theta,\varphi}(\mathbf{x}_0) \right] \tag{31}$$

$$= \mathbb{E}_{q(\mathbf{x}_0)} \left[ -\log \int \int \bar{p}(\mathbf{x}_0|\bar{\mathbf{z}}_0) \bar{p}_{\theta,\varphi}(\bar{\mathbf{z}}_{0:1-\Delta t}|\bar{\mathbf{z}}_1) \bar{p}(\bar{\mathbf{z}}_1|\mathbf{x}_1) q(\mathbf{x}_1) dz_{0:1} dx_1 \right] \tag{32}$$

$$= \mathbb{E}_{q(\mathbf{x}_0)} \left[ -\log \mathbb{E}_{\bar{q}(\mathbf{x}_1,\bar{\mathbf{z}}_{0:1}|\mathbf{x}_0)} \left[ \bar{p}(\mathbf{x}_0|\bar{\mathbf{z}}_0) \frac{\bar{p}_{\theta,\varphi}(\bar{\mathbf{z}}_{0:1-\Delta t}|\bar{\mathbf{z}}_1) \bar{p}(\bar{\mathbf{z}}_1|\mathbf{x}_1) q(\mathbf{x}_1)}{\bar{q}_\varphi(\bar{\mathbf{z}}_{0:1-\Delta t}|\bar{\mathbf{z}}_1,\mathbf{x}_0,\mathbf{x}_1) \bar{q}_\varphi(\bar{\mathbf{z}}_1|\mathbf{x}_0,\mathbf{x}_1) q(\mathbf{x}_1|\mathbf{x}_0)} \right] \right] \tag{33}$$

$$\leq \mathbb{E}_{q(\mathbf{x}_0,\mathbf{x}_1,\mathbf{z}_{0:1})} \left[ -\log \bar{p}(\mathbf{x}_0|\bar{\mathbf{z}}_0) \frac{\bar{p}_{\theta,\varphi}(\bar{\mathbf{z}}_{0:1-\Delta t}|\bar{\mathbf{z}}_1) \bar{p}(\bar{\mathbf{z}}_1|\mathbf{x}_1) q(\mathbf{x}_1)}{\bar{q}_\varphi(\bar{\mathbf{z}}_{0:1-\Delta t}|\bar{\mathbf{z}}_1,\mathbf{x}_0,\mathbf{x}_1) \bar{q}(\bar{\mathbf{z}}_1|\mathbf{x}_0,\mathbf{x}_1) q(\mathbf{x}_1|\mathbf{x}_0)} \right] \tag{34}$$

$$= \bar{\mathcal{L}}_{\text{rec}} + \bar{\mathcal{L}}_{\text{diff}} + \bar{\mathcal{L}}_{\text{prior}} + \text{const}, \quad \text{where} \tag{35}$$

$$\bar{\mathcal{L}}_{\text{rec}} = \mathbb{E}_{q(\mathbf{x}_0,\mathbf{z}_0)}\left[ -\log \bar{p}(\mathbf{x}_0|\bar{\mathbf{z}}_0) \right], \tag{36}$$

$$\bar{\mathcal{L}}_{\text{diff}} = \sum_{t=\frac{1}{T}}^{T} \mathbb{E}_{\bar{q}_\varphi(\mathbf{x}_0,\mathbf{x}_1,\bar{\mathbf{z}}_t)} \left[ D_{\text{KL}}\big(\bar{q}_\varphi(\bar{\mathbf{z}}_{t-\Delta t}|\bar{\mathbf{z}}_t,\mathbf{x}_0,\mathbf{x}_1) \| \bar{p}_{\theta,\varphi}(\bar{\mathbf{z}}_{t-\Delta t}|\bar{\mathbf{z}}_t)\big) \right], \tag{37}$$

$$\bar{\mathcal{L}}_{\text{prior}} = \mathbb{E}_{q(\mathbf{x}_0,\mathbf{x}_1)} \left[ D_{\text{KL}}\big(\bar{q}_\varphi(\bar{\mathbf{z}}_1|\mathbf{x}_0,\mathbf{x}_1) \| \bar{p}(\bar{\mathbf{z}}_1|\mathbf{x}_1)\big) \right], \tag{38}$$

$$\text{const} = \mathbb{E}_{q(\mathbf{x}_0)} \left[ D_{\text{KL}}\big(q(\mathbf{x}_1|\mathbf{x}_0) \| q(\mathbf{x}_1)\big) \right]. \tag{39}$$

Here, the reconstruction term $\mathcal{L}_{\text{rec}} = \bar{\mathcal{L}}_{\text{rec}}$ and the prior term $\mathcal{L}_{\text{prior}} = \bar{\mathcal{L}}_{\text{prior}}$ do not depend on discretization, and the constant term does not depend on model's parameters. For the diffusion term

$\bar{\mathcal{L}}_{\text{diff}}$ it is easy to see that in the case of NFBM it has the same structure as in NFDM. Therefore, taking the limit of the number of discretization steps $T$ the diffusion term $\bar{\mathcal{L}}_{\text{diff}}$ transforms to:

$$\mathcal{L}_{\text{diff}} = \mathbb{E}_{u(t)\bar{q}_\varphi(\mathbf{x}_0, \mathbf{x}_1, \bar{\mathbf{z}}_t)} \left[ \frac{1}{2g_\varphi^2(t)} \left\| \tilde{f}_\varphi^B(\bar{\mathbf{z}}_t, t, \mathbf{x}_0, \mathbf{x}_1) - \hat{f}_{\theta,\varphi}(\bar{\mathbf{z}}_t, t) \right\|_2^2 \right], \quad (40)$$

where $u(t)$ is a uniform distribution over a unit interval.

This gives us a variational bound on the negative log-likelihood of the NFBM, with a very similar structure as the NFDM objective. Additionally, for NFBM we also may omit optimization of $\mathcal{L}_{\text{rec}}$ and prior $\mathcal{L}_{\text{prior}}$ when $q_\varphi(\mathbf{z}_0|\mathbf{x}_0, \mathbf{x}_1) \approx \delta(\mathbf{x} - \mathbf{z}_0)$ and $q_\varphi(\mathbf{z}_1|\mathbf{x}_0, \mathbf{x}_1) \approx p(\mathbf{z}_1|\mathbf{x}_1)$.

## B  Details of NFDM

### B.1  Motivation of Generalized Forward Process

In this section, we provide an extended discussion on the motivation behind the learnable forward process.

From a theoretical perspective, diffusion models can be viewed as a hierarchical VAE, where the objective function of diffusion models is the variational bound on model log-likelihood. Therefore, diffusion models aim to match the joint distributions of data $\mathbf{x}$ and latent variables $\mathbf{z}_{0:1}$ (for simplicity, let's consider a discrete-time model with some $T$ time steps). However, as we know from [30, 45], the variational objective is equal to:

$$\underbrace{D_{\text{KL}}\big(q_\varphi(\mathbf{x}, \mathbf{z}_{0:1}) \| p_\theta(\mathbf{x}, \mathbf{z}_{0:1})\big)}_{\text{objective}} = \underbrace{D_{\text{KL}}\big(q(\mathbf{x}) \| p_\theta(\mathbf{x})\big)}_{\text{entropy + likelihood}} + \underbrace{D_{\text{KL}}\big(q_\varphi(\mathbf{z}_{0:1}|\mathbf{x}) \| p_\theta(\mathbf{z}_{0:1}|\mathbf{x})\big)}_{\text{posterior error}}. \quad (41)$$

Thus, by minimizing the variational objective in diffusion models, we address two aspects: the likelihood of the data and the posterior error. From this perspective, it is quite a natural choice to also optimize the variational distribution $q_\varphi(\mathbf{z}_{0:1}|\mathbf{x})$ to match the model's posterior distribution $p_\theta(\mathbf{z}_{0:1}|\mathbf{x})$.

In other words, by making the forward process learnable, we optimize not only the reverse process to match the forward process but also allow the forward process to match the reverse process. This approach simplifies the task of matching the data distribution by the reverse process.

### B.2  Calculation of the Time Derivative

As discussed in Section 3.1, the definition of forward process requires access to the time derivative of the function $F_\varphi$. This derivative is crucial for constructing a conditional ODE (see eq. (8)):

$$f_\varphi(\mathbf{z}_t, t, \mathbf{x}) = \left. \frac{\partial F_\varphi(\varepsilon, t, \mathbf{x})}{\partial t} \right|_{\varepsilon = F_\varphi^{-1}(\mathbf{z}_t, t, \mathbf{x})}. \quad (42)$$

First, let us clarify this notation: to find the ODE drift term $f_\varphi$ at point $\mathbf{z}_t$ and time $t$ conditionally on $\mathbf{x}$, we first apply the inverse function $F_\varphi^{-1}$ to identify $\varepsilon$, which corresponds to $\mathbf{z}_t$, $t$, and $\mathbf{x}$. Subsequently, we compute the partial derivative of $F_\varphi$ with respect to $t$ at $\varepsilon$ and $\mathbf{x}$.

For certain parameterizations of $F_\varphi$, the time derivative can be determined analytically. For instance, $F_\varphi$ associated with conventional diffusion models (see Appendix E.1) have time derivatives available in a closed form. However, in the general case, the time derivative is not readily available.

Fortunately, we can employ automatic differentiation tools like PyTorch [42] or JAX [5]. The naive approach to determining the time derivative involves running a backpropagation procedure for each output of $F_\varphi$ and reconstructing each coordinate of the time derivative. Unfortunately, this method is not scalable for high-dimensional $F_\varphi$.

Instead of backpropagation, which corresponds to a Vector Jacobian Product (VJP), we can use a forward differentiation mode, or Jacobian Vector Product (JVP). By fixing the inputs $\varepsilon$ and $\mathbf{x}$ of the function $F_\varphi$ and focusing solely on the input $t$, the Jacobian of $F_\varphi$ appears as a vector. Consequently, computing a JVP with a unit one-dimensional vector precisely produces the time derivative of $F_\varphi$.

This approach allows for scalable computation of time derivatives.

## B.3 Calculation of the Conditional Score Function

In Section 3.1, we define the forward process as a conditional SDE (eq. (9)):

$$d\mathbf{z}_t = \tilde{f}_\varphi^F(\mathbf{z}_t, t, \mathbf{x})dt + g_\varphi(t)d\mathbf{w}, \quad \text{where} \tag{43}$$

$$\tilde{f}_\varphi^F(\mathbf{z}_t, t, \mathbf{x}) = f_\varphi(\mathbf{z}_t, t, \mathbf{x}) + \frac{g_\varphi^2(t)}{2}\nabla_{\mathbf{z}_t} \log q_\varphi(\mathbf{z}_t|\mathbf{x}).$$

To construct this SDE, access to the conditional score function $\nabla_{\mathbf{z}_t} \log q_\varphi(\mathbf{z}_t|\mathbf{x})$ is required. This section discusses methods for calculating this score function.

First, consider the case where $F_\varphi$ is linear in $\varepsilon$ (further details on this parameterization are discussed in Appendix B.5):

$$F_\varphi(\varepsilon, t, \mathbf{x}) = \mu_\varphi(\mathbf{x}, t) + \varepsilon\sigma_\varphi(\mathbf{x}, t), \tag{44}$$

where $\mu_\varphi : \mathbb{R}^D \times [0,1] \mapsto \mathbb{R}^D$ and $\sigma_\varphi : \mathbb{R}^d \times [0,1] \mapsto \mathbb{R}^D$, with the product $\varepsilon\sigma_\varphi(\mathbf{x}, t)$ being element-wise. Consequently, the marginal probability densities of the forward process are Gaussian:

$$q_\varphi(\mathbf{z}_t|\mathbf{x}) = \mathcal{N}(\mathbf{z}_t; \mu_\varphi(\mathbf{x}, t), \sigma_\varphi^2(\mathbf{x}, t)). \tag{45}$$

In the case of Gaussian distributions, the score function can be easily found as:

$$\nabla_{\mathbf{z}_t} \log q_\varphi(\mathbf{z}_t|\mathbf{x}) = \frac{\mu_\varphi(\mathbf{x}, t) - \mathbf{x}}{\sigma_\varphi^2(\mathbf{x}, t)} = \left. -\frac{\varepsilon}{\sigma_\varphi(\mathbf{x}, t)} \right|_{\varepsilon = F_\varphi^{-1}(\mathbf{z}_t, t, \mathbf{x})}. \tag{46}$$

Now, consider the more general case where the distribution $q_\varphi(\mathbf{z}_t|\mathbf{x})$ is defined implicitly through an invertible transformation $F_\varphi$ of the random variable $\varepsilon \sim q(\varepsilon)$ into $\mathbf{z}_t$. To find the score function, we apply the change of variable formula:

$$\nabla_{\mathbf{z}_t} \log q_\varphi(\mathbf{z}_t|\mathbf{x}) = \nabla_{\mathbf{z}_t} \left[ \log q(\varepsilon) \Big|_{\varepsilon = F_\varphi^{-1}(\mathbf{z}_t, t, \mathbf{x})} + \log \left| \frac{\partial F_\varphi^{-1}(\mathbf{z}_t, t, \mathbf{x})}{\partial \mathbf{z}_t} \right| \right]. \tag{47}$$

The first part of the equation is the log-density of the noise distribution, which is straightforward to calculate. The second part is the log-determinant of the Jacobian matrix of the inverse transformation, $\log |J_F^{-1}|$. If we have access to $\log |J_F^{-1}|$, we can calculate $\log q_\varphi(\mathbf{z}_t|\mathbf{x})$ using the change of variable formula and subsequently compute the gradient using automatic differentiation tools such as PyTorch [42] or JAX [5].

The log-determinant $\log |J_F^{-1}|$ can be easily calculated when $F_\varphi$ is linear in $\varepsilon$, a case already considered above. For nonlinear, low-dimensional $F_\varphi$, the Jacobian $J_F^{-1}$ can be computed using $\mathcal{O}(D)$ backpropagations, and the log-determinant subsequently calculated. However, this approach is not scalable for high-dimensional cases. It is challenging to compute the log-determinant $\log |J_F^{-1}|$ in general high-dimensional cases, posing a limitation to the NFDM framework as it restricts the parameterization of the forward process to functions with accessible log-determinants.

Nevertheless, the forward process can be parameterized with functions $F_\varphi$ that inherently provide access to the log-determinant $\log |J_F^{-1}|$ by design. For example, normalizing flow architectures [14, 29] facilitate the construction of forward processes with non-Gaussian, learnable distributions $q_\varphi(\mathbf{z}_t|\mathbf{x})$.

## B.4 Stochastic and Deterministic Sampling

In Section 3.2, we define the reverse process as the following SDE (see eq. (11)):

$$d\mathbf{z}_t = \hat{f}_{\theta,\varphi}(\mathbf{z}_t, t)dt + g_\varphi(t)d\bar{\mathbf{w}}, \quad \text{where} \quad \hat{f}_{\theta,\varphi}(\mathbf{z}_t, t) = \tilde{f}_\varphi^B(\mathbf{z}_t, t, \hat{\mathbf{x}}_\theta(\mathbf{z}_t, t)). \tag{48}$$

Let us now rewrite this equation, substituting the definition of $\tilde{f}_\varphi^B$ from eq. (10):

$$d\mathbf{z}_t = \left[ f_\varphi(\mathbf{z}_t, t, \hat{\mathbf{x}}_\theta(\mathbf{z}_t, t)) - \frac{g_\varphi^2(t)}{2}\nabla_{\mathbf{z}_t} \log q_\varphi(\mathbf{z}_t|\hat{\mathbf{x}}_\theta(\mathbf{z}_t, t)) \right] dt + g_\varphi(t)d\bar{\mathbf{w}}. \tag{49}$$

This expression represents the full form of the generative SDE, for which we derive the objective in Appendix A.1. After training, to sample data points from the models, we can simulate this SDE as outlined in eq. (49) and depicted in Algorithm 2.

However, as evident from eq. (49), the generative process has a very simple dependence on $g_\varphi(t)$, which determines the level of stochasticity. Therefore, in practice, we can adjust the stochasticity level by modifying the function $g_\varphi(t)$.

In the extreme case where $g_\varphi(t) \equiv 0$, the generative SDE completely loses its stochasticity and becomes an ODE:

$$d\mathbf{z}_t = f_\varphi\big(\mathbf{z}_t, t, \hat{\mathbf{x}}_\theta(\mathbf{z}_t, t)\big)dt. \tag{50}$$

This ODE allows for an interpretation of the model as a Continuous Normalizing Flow (CNF) [6, 17], consequently enabling deterministic sampling from a pre-trained model and likelihood estimation.

## B.5 Parameterization of the Forward Process

The parameterization of the variance function $g_\varphi(t)$ (eq. (9)) and the data point predictor $\hat{\mathbf{x}}_\theta(\mathbf{z}_t, t)$ (eq. (11)) is straightforward. In our experiments, we parameterize them directly using neural networks. However, parameterizing the transformation $F_\varphi$ (eq. (7)) is more complex. NFDM requires $F_\varphi$ to be differentiable and invertible with respect to $\varepsilon$, as well as provide access to the log-determinant of the Jacobian matrix (see Appendix B.3). We propose two parameterizations for the forward process of NFDM.

First, we propose a parameterization of $F_\varphi$ that is linear in $\varepsilon$:

$$F_\varphi(\varepsilon, t, \mathbf{x}) = \mu_\varphi(\mathbf{x}, t) + \varepsilon\sigma_\varphi(\mathbf{x}, t), \quad \text{where} \tag{51}$$

$$\mu_\varphi(\mathbf{x}, t) = (1 - t)\mathbf{x} + t(1 - t)\bar{\mu}_\varphi(\mathbf{x}, t), \tag{52}$$

$$\sigma_\varphi(\mathbf{x}, t) = \delta^{1-t} \left(\bar{\sigma}_\varphi(\mathbf{x}, t)\right)^{t(1-t)}. \tag{53}$$

Here, $\bar{\mu}_\varphi : \mathbb{R}^D \times [0, 1] \mapsto \mathbb{R}^D$ and $\bar{\sigma}_\varphi : \mathbb{R}^D \times [0, 1] \mapsto \mathbb{R}^D_+$, with $\delta$ as a constant. This parameterization models $q(\mathbf{z}_t|\mathbf{x})$ as a conditional Gaussian distribution:

$$q_\varphi(\mathbf{z}_t|\mathbf{x}) = \mathcal{N}(\mathbf{z}_t; \mu_\varphi(\mathbf{x}, t), \sigma_\varphi^2(\mathbf{x}, t)). \tag{54}$$

In our experiments, we set $\delta^2 = 10^{-4}$. If $\bar{\mu}_\varphi$ is differentiable and $\bar{\sigma}_\varphi$ is differentiable and non-negative, this parameterization meets the requirements on $F_\varphi$.

The functions $\mu_\varphi(\mathbf{x}, t)$ and $\sigma_\varphi(\mathbf{x}, t)$ are specifically designed such that at $t = 0$, they equal $\mu_\varphi(\mathbf{x}, 0) = \mathbf{x}$ and $\sigma_\varphi(\mathbf{x}, t) = \delta$, and at $t = 1$, they become $\mu_\varphi(\mathbf{x}, 1) = 0$ and $\sigma_\varphi(\mathbf{x}, t) = I$. At intermediate times $0 < t < 1$, these functions can take arbitrary values, thereby ensuring that $q(\mathbf{z}_0|\mathbf{x}) = \mathcal{N}(\mathbf{z}_0; \mathbf{x}, \delta^2 I)$ and $q(\mathbf{z}_1|\mathbf{x}) = \mathcal{N}(\mathbf{z}_1; 0, I)$, with arbitrary Gaussian distributions at intermediate steps. These restrictions are not necessary for $F_\varphi$, however, it eliminates the need to optimize the reconstruction loss $\mathcal{L}_{\text{rec}}$ and the prior loss $\mathcal{L}_{\text{prior}}$, as they become independent of the parameters $\varphi$ and $\theta$ (see Appendix A.1).

Secondly, we propose a more general parameterization:

$$F_\varphi(\varepsilon, t, \mathbf{x}) = (1 - t)\mathbf{x} + \big(\delta + (1 - \delta)t\big)\bar{F}_\varphi(\varepsilon, t, (1 - t)\mathbf{x}), \tag{55}$$

where $\bar{F}_\varphi : \mathbb{R}^D \times [0, 1] \times \mathbb{R}^D \mapsto \mathbb{R}^D$ is an invertible and differentiable function that allows access to the log-determinant of the Jacobian matrix. If $\bar{F}_\varphi$ is sufficiently flexible (e.g., a neural network with a normalizing flow architecture), it can parameterize non-Gaussian distributions $q_\varphi(\mathbf{z}_t|\mathbf{x})$.

The additive term $(1 - t)\mathbf{x}$ provides an inductive bias that shifts the mean of the distribution $q_\varphi(\mathbf{z}_t|\mathbf{x})$ towards $\mathbf{x}$ when $t$ is near 0. The linear function $\delta + (1 - \delta)t$ scales the distribution from a factor of $\delta$ at $t = 0$ to 1 at $t = 1$, also imparting an inductive bias for increasing variance over time. The scaling $(1 - t)\mathbf{x}$ in the third argument of $\bar{F}_\varphi$ ensures that $q_\varphi(\mathbf{z}_1|\mathbf{x})$ at $t = 1$ does not depend on $\mathbf{x}$, allowing the reverse process to start from $p(\mathbf{z}_1) = q_\varphi(\mathbf{z}_1|\mathbf{0})$ and thereby guaranteeing matching of prior distributions between the forward and reverse processes. Consequently, this eliminates the need for optimizing the prior loss $\mathcal{L}_{\text{prior}}$. However, unlike the previous parameterization that guarantees $q(\mathbf{z}_0|\mathbf{x}) = \mathcal{N}(\mathbf{z}_0; \mathbf{x}, \delta^2 I)$, in this case, there are no guarantees on the form of $q(\mathbf{z}_0|\mathbf{x})$, necessitating the optimization of the reconstruction loss $\mathcal{L}_{\text{rec}}$ with respect to parameters $\varphi$ (see Appendix A.1).

Importantly, even the Gaussian parameterization of $F_\varphi$ in eq. (51) offers a significantly more flexible forward process compared to prior works. Although linear in $\varepsilon$, it may exhibit complex non-linear dependencies on $\mathbf{x}$ and $t$.

Furthermore, we emphasize that the parameterizations of $F_\varphi$, $g_\varphi(t)$, and $\hat{\mathbf{x}}_\theta(\mathbf{z}_t, t)$ discussed here are not the only possible ones and may be sub-optimal. For the purposes of this paper, we maintain a simple setup and leave the exploration of more effective parameterizations for future research.

## C Details of NFBM

### C.1 Parameterization of the Forward Process

To parameterize the forward process of NFBM, we adopt the Gaussian parameterization discussed for NFDM in Appendix B.5:

$$F_\varphi(\varepsilon, t, \mathbf{x}_0, \mathbf{x}_1) = \mu_\varphi(\mathbf{x}_0, \mathbf{x}_1, t) + \varepsilon\sigma_\varphi(\mathbf{x}_0, \mathbf{x}_1, t), \quad \text{where} \tag{56}$$

$$\mu_\varphi(\mathbf{x}_0, \mathbf{x}_1, t) = (1-t)\mathbf{x}_0 + t\mathbf{x}_1 + t(1-t)\bar{\mu}_\varphi(\mathbf{x}_0, \mathbf{x}_1, t), \tag{57}$$

$$\sigma_\varphi(\mathbf{x}_0, \mathbf{x}_1, t) = \delta\left(\bar{\sigma}_\varphi(\mathbf{x}_0, \mathbf{x}_1, t)\right)^{t(1-t)}. \tag{58}$$

Here, $\bar{\mu}_\varphi : \mathbb{R}^D \times \mathbb{R}^D \times [0,1] \mapsto \mathbb{R}^D$ and $\bar{\sigma}_\varphi : \mathbb{R}^D \times \mathbb{R}^D \times [0,1] \mapsto \mathbb{R}^D_+$, with $\delta^2 = 10^{-4}$ in our experiments. Similar to the NFDM, this parameterization models $q(\mathbf{z}_t|\mathbf{x})$ as a conditional Gaussian distribution:

$$q_\varphi(\mathbf{z}_t|\mathbf{x}_0, \mathbf{x}_1) = \mathcal{N}(\mathbf{z}_t; \mu_\varphi(\mathbf{x}_0, \mathbf{x}_1, t), \sigma_\varphi^2(\mathbf{x}_0, \mathbf{x}_1, t)). \tag{59}$$

In contrast to the parameterization of NFDM, here $\mu_\varphi(\mathbf{x}, t)$ and $\sigma_\varphi(\mathbf{x}, t)$ are designed to ensure that $q(\mathbf{z}_0|\mathbf{x}_0, \mathbf{x}_1) = \mathcal{N}(\mathbf{z}_0; \mathbf{x}_0, \delta^2 I)$ and $q(\mathbf{z}_1|\mathbf{x}_0, \mathbf{x}_1) = \mathcal{N}(\mathbf{z}_1; \mathbf{x}_1, \delta^2 I)$. At intermediate times, $0 < t < 1$, $q(\mathbf{z}_t|\mathbf{x}_0, \mathbf{x}_1)$ may still be an arbitrary Gaussian distribution. Therefore, with this parameterization of NFBM, we do not need to optimize the reconstruction loss $\mathcal{L}_{\text{rec}}$ and the prior loss $\mathcal{L}_{\text{prior}}$, as they are independent of the parameters $\varphi$ and $\theta$ (see Appendix A.2).

As with NFDM, the parameterization of NFBM that we provide in this section represents a simple design choice. We believe there are more efficient ways to parameterize the forward process of NFBM. However, in this paper, we want to focus on presenting the framework itself, rather than dwelling on design choices.

### C.2 Parameterization of the Reverse Process

In Section 4, we define the reverse process of NFBM by the following SDE:

$$d\mathbf{z}_t = \hat{f}_{\theta,\varphi}(\mathbf{z}_t, t)dt + g_\varphi(t)d\bar{\mathbf{w}}, \quad \text{where} \quad \hat{f}_{\theta,\varphi}(\mathbf{z}_t, t) = \tilde{f}_\varphi^B\left(\mathbf{z}_t, t, \hat{\mathbf{x}}_\theta^{0,1}(\mathbf{z}_t, t)\right). \tag{60}$$

Here, the function $\hat{\mathbf{x}}_\theta^{0,1}(\mathbf{z}_t, t)$ predicts both $\mathbf{x}_0$ and $\mathbf{x}_1$. Assuming $q_\varphi(\mathbf{z}_0|\mathbf{x}_0, \mathbf{x}_1) \approx \delta(\mathbf{x} - \mathbf{z}_0)$ and $q_\varphi(\mathbf{z}_1|\mathbf{x}_0, \mathbf{x}_1) \approx p(\mathbf{z}_1|\mathbf{x}_1)$, this formulation leads to the objective (see Appendix C.1):

$$\mathcal{L} = \mathbb{E}_{u(t)q(\mathbf{x}_0, \mathbf{x}_1)q_\varphi(\mathbf{z}_t|\mathbf{x}_0, \mathbf{x}_1)}\left[\frac{1}{2g_\varphi^2(t)}\left\|\tilde{f}_\varphi^B(\mathbf{z}_t, t, \mathbf{x}_0, \mathbf{x}_1) - \hat{f}_{\theta,\varphi}(\mathbf{z}_t, t)\right\|_2^2\right]. \tag{61}$$

As previously mentioned, alternatively, we could define the reverse process as being fully conditional on the initial data point $\mathbf{x}_1$. This leads to the following SDE:

$$d\mathbf{z}_t = \hat{f}_{\theta,\varphi}(\mathbf{z}_t, t)dt + g_\varphi(t)d\bar{\mathbf{w}}, \quad \text{where} \quad \hat{f}_{\theta,\varphi}(\mathbf{z}_t, t) = \tilde{f}_\varphi^B\left(\mathbf{z}_t, t, \hat{\mathbf{x}}_\theta^0(\mathbf{z}_t, t, \mathbf{x}_1), \mathbf{x}_1\right). \tag{62}$$

This approach is akin to classifier-free guidance, where each step of the generative process depends on conditioning $\mathbf{x}_1$. Under the same assumptions—that $q_\varphi(\mathbf{z}_0|\mathbf{x}_0, \mathbf{x}_1) \approx \delta(\mathbf{x} - \mathbf{z}_0)$ and $q_\varphi(\mathbf{z}_1|\mathbf{x}_0, \mathbf{x}_1) \approx p(\mathbf{z}_1|\mathbf{x}_1)$—this definition leads to a slightly different objective (the derivation is not provided here but can be easily inferred by following the steps described in Appendix A.2 for the previous definition):

$$\mathcal{L} = \mathbb{E}_{u(t)q(\mathbf{x}_0, \mathbf{x}_1)q_\varphi(\mathbf{z}_t|\mathbf{x}_0, \mathbf{x}_1)}\left[\frac{1}{2g_\varphi^2(t)}\left\|\tilde{f}_\varphi^B(\mathbf{z}_t, t, \mathbf{x}_0, \mathbf{x}_1) - \hat{f}_{\theta,\varphi}(\mathbf{z}_t, t, \mathbf{x}_1)\right\|_2^2\right]. \tag{63}$$

Intuitively, the difference between these two definitions implies that the first reverse process, at a point $\mathbf{z}_t$ at time $t$, follows the average trajectory of forward processes passing through $\mathbf{z}_t$. Conversely, the second reverse process tracks the average trajectory of forward processes that pass through $\mathbf{z}_t$ and terminate at $\mathbf{z}_1$.

The second approach may be beneficial when dealing with joint data distributions $q(\mathbf{x}_0, \mathbf{x}_1)$. However, in this paper, as we work with unpaired distributions $q(\mathbf{x}_0)q(\mathbf{x}_1)$, we have opted for the first approach in Section 4.

# D   NFDM with restrictions

## D.1   Additional Discussion of NFDM with Restrictions

In Section 5, we discussed training the NFDM with restrictions on the reverse process. This section aims to further explore the training under restrictions and to examine the limitations of this technique.

Typically, there is an infinite number of forward and reverse processes that correspond to each other. So how to choose between them? The standard NFDM, as described in Section 3, learns just one such pair of processes. However, the flexibility of NFDM opens up the possibility of choosing specific pairs of processes depending on the task.

Suppose we aim to learn a reverse process with some specific properties. To achieve this, we must impose certain restrictions on the reverse process. One approach is to encode these restrictions directly in the parameterization of the reverse process. Alternatively, we can introduce penalties on the reverse process that would penalize deviations from the desired properties. In both cases, we expect the forward process to adapt and align with the reverse process. Consequently, the reverse process will match the forward process and, as a result, the data distribution, while possessing the desired properties.

In Section 5, we discussed the restriction of the reverse process to have straight deterministic generative trajectories. However, we may apply the same strategy to ensure arbitrary properties. We believe that the NFDM framework may help construct various new generative dynamics for a variety of tasks and domains.

Nevertheless, training the NFDM with restrictions has limitations. The primary concern is the feasibility of these restrictions. For instance, imposing overly stringent constraints on the reverse process (such as fixing its drift term $\hat{f} \equiv 0$) will render the forward process incapable of adapting, regardless of its flexibility. Similarly, unattainable penalties, such as those on the squared norm of the drift term $\|\hat{f}_{\theta,\varphi}\|_2^2$, will lead to biased solutions, as it is impossible to match two processes when $\|\hat{f}_{\theta,\varphi}\|_2^2 \equiv 0$.

Additionally, even when a restriction is feasible, if the model parameterization is too simplistic, it may not be able to match processes and satisfy the restriction, leading to bias. However, in our experiments, we find that even the Gaussian parameterization of the forward process (see Appendix B.5) is flexible enough in practice to successfully learn various restrictions, such as straight generative trajectories.

## D.2   Details of NFDM with Curvature Penalty

In Section 5, we suggest penalizing the curvature of deterministic generative trajectories specifically with the $\mathcal{L}_{\mathrm{crv}}$ (eq. (16)) penalty.

To calculate the curvature penalty $\mathcal{L}_{\mathrm{crv}}$ (eq. (16)), we proceed as follows. First, we use the chain rule to rewrite the time derivative:

$$\frac{d\hat{f}_{\theta,\varphi}(\mathbf{z}_t, t)}{dt} = \frac{\partial \hat{f}_{\theta,\varphi}(\mathbf{z}_t, t)}{\partial \mathbf{z}_t} \frac{\partial \mathbf{z}_t}{\partial t} + \frac{\partial \hat{f}_{\theta,\varphi}(\mathbf{z}_t, t)}{\partial t} \tag{64}$$

$$= \frac{\partial \hat{f}_{\theta,\varphi}(\mathbf{z}_t, t)}{\partial \mathbf{z}_t} \hat{f}_{\theta,\varphi}(\mathbf{z}_t, t) + \frac{\partial \hat{f}_{\theta,\varphi}(\mathbf{z}_t, t)}{\partial t}. \tag{65}$$

The second term in eq. (65) is the time derivative of a function. As discussed in appendix B.2, the time derivatives can be determined as a Jacobian Vector Product (JVP) with a one-dimensional unit vector.

Notably, the first term in eq. (65) is also a JVP. Therefore, we can combine these two operations. For this purpose, we define a $\mathbb{R}^{D+1}$ dimensional adjoint vector $v$:

$$v = \begin{bmatrix} \hat{f}_{\theta,\varphi}(\mathbf{z}_t, t) \\ 1 \end{bmatrix}. \tag{66}$$

Consequently, $\mathcal{L}_{\text{crv}}$ can be efficiently computed as the JVP of $\hat{f}_{\theta,\varphi}(\mathbf{z}_t, t)$ with the vector $v$.

Additionally, we would like to note a couple of aspects of learning the NFDM with a curvature penalty. First, unlike the regular NFDM, the curvature penalty necessitates that $F_{\varphi}$ be twice differentiable with respect to time. Second, although we parameterize $\hat{f}_{\theta,\varphi}$ (eq. (11)) through $\tilde{f}_{\varphi}^B$ (eq. (10)), $\mathcal{L}_{\text{crv}}$ influences the forward process but does not penalize the curvature of the conditional forward trajectories (see illustrations in Appendix G.1).

# E   Connections of NFDM with Other Approaches

In this section we continue discussing connections between NFDM and related works.

## E.1   Special Cases of NFDM

Many existing works define the forward process in diffusion models as either fixed or simply parameterized processes. These processes can be considered special cases of NFDM. In this section, we review some of these approaches.

**Soft Diffusion**. [9] consider the case of a fixed forward processes $q(\mathbf{z}_t|\mathbf{x}) = \mathcal{N}(\mathbf{z}_t; C_t\mathbf{x}, \sigma_t I)$, which can be reparameterized as $F(\varepsilon, t, \mathbf{x}) = C_t\mathbf{x} + \sigma_t\varepsilon$. Such distributions include, for example, combinations of blurring and the injection of Gaussian noise.

**Star-Shaped Diffusion**. [41] extended conventional diffusion models to include distributions from the exponential family. Although Star-Shaped Diffusion is a discrete-time approach and does not directly correspond to NFDM, the latter can work with exponential family distributions through reparameterization functions. For instance, for some continuous one-dimensional distribution $q(\mathbf{z}_t|\mathbf{x})$, NFDM could use $F(\varepsilon, t, \mathbf{x}) = a(b(\varepsilon), t, \mathbf{x})$, where $a$ is the inverse Cumulative Distribution Function (CDF) of $q(\mathbf{z}_t|\mathbf{x})$, and $b$ is the CDF of a unit Gaussian.

**Variational Diffusion Models**. [28] proposed forward conditional distributions $q_{\varphi}(\mathbf{z}_t|\mathbf{x})$ as $\mathcal{N}(\mathbf{z}_t; \alpha_{\varphi}(t)\mathbf{x}, \sigma_{\varphi}^2(t)I)$ with learnable parameters $\varphi$. In the context of NFDM, this can be realized by $F_{\varphi}(\varepsilon, t, \mathbf{x}) = \alpha_{\varphi}(t)\mathbf{x} + \sigma_{\varphi}(t)\varepsilon$.

**LSGM**. [66] suggest an alternative approach for parameterizing the forward process, proposing diffusion in a latent space of a VAE. Therefore, the forward process is characterized by a distribution $q_{\varphi}(\mathbf{z}_t|\mathbf{x}) = \mathcal{N}(\mathbf{z}_t; \alpha_t E_{\varphi}(\mathbf{x}), \sigma_t^2 I)$, where $E_{\varphi}$ is the encoder of the VAE. To parameterize the same forward process with NFDM, one could use $F_{\varphi}(\varepsilon, t, \mathbf{x}) = \alpha_t E_{\varphi}(\mathbf{x}) + \sigma_t\varepsilon$. To align the reverse process, the reconstruction distribution should be $p(\mathbf{x}|\mathbf{z}_0) = \mathcal{N}(\mathbf{x}; D_{\varphi}(\mathbf{z}_0), \delta^2 I)$, where $D_{\varphi}$ is VAE's decoder.

**ShiftDDPM**. [74] proposed learning a function that shifts the mean and covariance of $q_{\phi}(\mathbf{z}_t|\mathbf{x})$ by conditional information. In our work, we do not focus on conditional generation. However, we believe that the NFDM framework can be readily adapted for conditional generation by incorporating the conditioning variable $\mathbf{c}$ into both the transformation $F_{\phi}(\varepsilon, t, \mathbf{x}, \mathbf{c})$ and the predictor $\hat{\mathbf{x}}(\mathbf{z}_t, t, \mathbf{c})$.

**NDM** and **DiffEnc**. [4, 40] proposed a more general forward process, $q_{\varphi}(\mathbf{z}_t|\mathbf{x}) = \mathcal{N}(\mathbf{z}_t; \alpha_t f_{\varphi}(\mathbf{x}, t), \sigma_t^2 I)$, which, unlike LSGM, transforms $\mathbf{x}$ in a time-dependent manner. This forward process can also be described in terms of NFDM as the special case $F_{\varphi}(\varepsilon, t, \mathbf{x}) = \alpha_t f_{\varphi}(\mathbf{x}, t) + \sigma_t\varepsilon$.

**Curvature minimization**. [32] suggested learning the distribution $q_{\varphi}(\mathbf{z}_1|\mathbf{x})$ of the forward process at time step $t = 1$ as a conditional Gaussian. They parameterize the forward process via linear interpolation between $\mathbf{x}$ and $\mathbf{z}_1$. Within the NFDM framework, we can express this as $F_{\varphi}(\varepsilon, t, \mathbf{x}) = (1-t)\mathbf{x} + t(\mu_{\varphi}(\mathbf{x}) + \sigma_{\varphi}(\mathbf{x})\varepsilon)$. Subsequently, the authors train their model by optimizing the Flow Matching objective in conjunction with minimizing the Kullback–Leibler (KL) divergence between

$q_\varphi(\mathbf{z}_1|\mathbf{x})$ and $p(\mathbf{z})$. This objective corresponds to minimization of diffusion term $\mathcal{L}_{\text{diff}}$ and prior term $\mathcal{L}_{\text{prior}}$ of NFDM objective (see Appendix A.1) for the forward process proposed by [32].

## E.2 Compatible and Orthogonal Approaches

Despite the simulation-free nature of the training procedure, diffusion models still necessitate full reverse process simulations for sample generation, leading to slow and computationally expensive inference. To address this, in an orthogonal line of works alternative sampling methods have been studied, such as deterministic sampling in discrete [57] and continuous time [61] and novel numerical solvers [62, 35, 38, 53]. Additionally, distillation techniques have been applied to both discrete [50] and continuous time models [58, 75, 72] to enhance sampling speed, albeit at the cost of training additional models. Most of these approaches are compatible with NFDM and may be adapted for further gains.

The significance of deterministic trajectories in efficient sampling was highlighted by [24]. At the same time, building on diffusion model concepts, [33, 37, 2] introduced simulation-free methods for learning deterministic generative dynamics. Based on these ideas, [37] and [36] proposed distillation procedures to straighten deterministic generative trajectories. Since these methods are based on distillation, we consider them orthogonal to NFDM.

## E.3 Other Related Works

**Schrödinger Bridge models**. Recent studies [73, 10, 54] have explored generative models based on Schrödinger Bridge theory and finite-time diffusion constructions. Unlike NFDM, these models deviate from the simulation-free paradigm by necessitating the full simulation of stochastic processes for training, which significantly increases their computational cost. Moreover, these studies typically aim to address the optimal stochastic control problem, which is outside the scope of our work.

Nevertheless, it is important to highlight that within this research domain, some authors have proposed advanced techniques for parameterizing learnable conditional processes. For instance, in [34], the authors introduced the use of splines to parameterize the conditional process, thereby constructing a more flexible and efficient model. This parameterization technique can be seen as a specific instantiation of the parameterization procedure proposed by the NFDM framework.

**Minibatch optimal transport**. [44] and [64] proposed to construct the forward process with optimal data-noise couplings to learn straighter generative trajectories. While this method is theoretically justified, it relies on estimating an optimal coupling over minibatches of the entire dataset, which, for large datasets, may become uninformative as to the true optimal transport coupling. In contrast, NFDM-OT directly penalizes curvature of the generative trajectories.

**Stochastic Interpolants**. In this section, we explore the connections between NFDM and Stochastic Interpolants, as proposed by [1]. Both NFDM and Stochastic Interpolants introduce a more general family of forward processes through the use of a reparameterization function. Although we acknowledge the relevance of Stochastic Interpolants to our work, there are notable differences between the two approaches.

First, our methodology involves parameterizing the reverse process by substituting the prediction of $\mathbf{x}$ into the forward process, whereas Stochastic Interpolants necessitate learning two separate functions for the reverse process: the velocity field and the score function.

Second, we present NFDM as a framework that enables both the pre-specification and learning of the forward process, in contrast to Stochastic Interpolants, which are derived under the assumption of a fixed forward process. Consequently, the objectives utilized by Stochastic Interpolants do not support the incorporation of learnable parameters for the forward process. Furthermore, these objectives are tailored towards learning the velocity field and the score function, rather than optimizing likelihood, as is the case with NFDM.

In their practical applications, Stochastic Interpolants are demonstrated with a simple parameterizations of the forward process. They propose a theoretical framework for learning the forward process that would result in an reverse process characterized by dynamical optimal transport. However, this approach is contingent upon solving a high-dimensional min-max optimization problem, for

which they do not provide experimental results. Moreover, their work does not clearly articulate how Stochastic Interpolants might be applied to learning other generative dynamics.

In contrast, NFDM introduces a more generic method for learning generative dynamics. Moreover, when NFDM is learned with restrictions or penalties (see Section 5), it remains within a min-min optimization paradigm, which can be addressed more efficiently.

### E.4 Connections of NFDM Objective

In this section, we delve into the details of the NFDM's objective function (eq. (12)). Specifically we will consider the diffusion term of the objective $\mathcal{L}_{\text{diff}}$ (see Appendix A.1).

First, let's unpack the $\mathcal{L}_{\text{diff}}$ by substituting the definitions of $\tilde{f}_\varphi^B$ and $\hat{f}_{\theta,\varphi}$:

$$\mathcal{L}_{\text{diff}} = \mathop{\mathbb{E}}_{u(t)q_\varphi(\mathbf{x},\mathbf{z}_t)} \left[ \frac{1}{2g_\varphi^2(t)} \left\| \tilde{f}_\varphi^B(\mathbf{z}_t, t, \mathbf{x}) - \hat{f}_{\theta,\varphi}(\mathbf{z}_t, t) \right\|_2^2 \right] \tag{67}$$

$$= \mathop{\mathbb{E}}_{u(t)q_\varphi(\mathbf{x},\mathbf{z}_t)} \left[ \frac{1}{2g_\varphi^2(t)} \left\| \underbrace{\left( f_\varphi(\mathbf{z}_t, t, \mathbf{x}) - f_\varphi(\mathbf{z}_t, t, \hat{\mathbf{x}}_\theta(\mathbf{z}_t, t)) \right)}_{\text{Flow Matching term}} + \right. \right. \tag{68}$$

$$\left. \left. \frac{g_\varphi^2(t)}{2} \underbrace{\left( \nabla_{\mathbf{z}_t} \log q_\varphi(\mathbf{z}_t | \hat{\mathbf{x}}_\theta(\mathbf{z}_t, t)) - \nabla_{\mathbf{z}_t} \log q_\varphi(\mathbf{z}_t | \mathbf{x}) \right)}_{\text{Score Matching term}} \right\|_2^2 \right].$$

This formulation clearly delineates two components of the objective: the first calculates the difference between the ODE drift terms, and the second calculates the difference between the score functions. Moreover, this expression highlights the role of $g_\varphi(t)$. When $g_\varphi(t)$ is small, the forward and reverse processes exhibit smoother trajectories, and the objective is dominated by the first term. Conversely, when $g_\varphi(t)$ is large, the processes exhibit more stochastic trajectories, and the objective is dominated by the second term.

Crucially, in the extreme scenarios where $g_\varphi(t)$ approaches either 0 or $\infty$, the diffusion loss $\mathcal{L}_{\text{diff}}$ corresponds to either a reweighted Flow Matching loss [33] or a reweighted Score Matching loss [68], respectively:

$$\lim_{g_\varphi(t)\to 0} \mathcal{L}_{\text{diff}} = \lim_{g_\varphi(t)\to 0} \mathbb{E}_{u(t)q_\varphi(\mathbf{x},\mathbf{z}_t)} \left[ \frac{1}{2g_\varphi^2(t)} \left\| f_\varphi(\mathbf{z}_t, t, \mathbf{x}) - f_\varphi(\mathbf{z}_t, t, \hat{\mathbf{x}}_\theta(\mathbf{z}_t, t)) \right\|_2^2 \right], \tag{69}$$

$$\lim_{g_\varphi(t)\to\infty} \mathcal{L}_{\text{diff}} = \lim_{g_\varphi(t)\to\infty} \mathbb{E}_{u(t)q_\varphi(\mathbf{x},\mathbf{z}_t)} \left[ \frac{g_\varphi^2(t)}{8} \left\| \nabla_{\mathbf{z}_t} \log q_\varphi(\mathbf{z}_t | \hat{\mathbf{x}}_\theta(\mathbf{z}_t, t)) - \nabla_{\mathbf{z}_t} \log q_\varphi(\mathbf{z}_t | \mathbf{x}) \right\|_2^2 \right] \tag{70}$$

This connection highlights the importance of optimizing the full objective in eq. (67), which contains both Flow Matching and Score Matching components. Simplifying the objective to include only one component renders it intractable from a variational perspective. Specifically, when $g_\phi(t)$ approaches either 0 or $\infty$, the objective, a variational bound on likelihood, becomes infinitely large.

There are two primary reasons why many popular diffusion models are trained using just one of these components.

Firstly, many diffusion models employ a simple linear parameterization of the forward process. For these models, both the conditional vector field $f_\varphi(\mathbf{z}_t, t, \mathbf{x})$ and the conditional score function $\nabla_{\mathbf{z}_t} \log q_\varphi(\mathbf{z}_t | \mathbf{x})$ are merely linear combinations of $\mathbf{z}_t$ and $\mathbf{x}$. As [27] demonstrated, for such models, Flow Matching and Score Matching can be viewed as reweighted ELBO. However, for more general non-linear forward process in NFDM, optimizing Flow Matching and Score Matching objectives is not equivalent.

Secondly, in other models, the Flow Matching and Score Matching objectives are derived under the assumption that the forward process is fixed. Without this assumption, we cannot derive either

Table 4: Training hyper-parameters.

| | CIFAR-10 | ImageNet 32 | ImageNet 64 | AFHQ 64 |
|---|---|---|---|---|
| Channels | 256 | 256 | 192 | 192 |
| Depth | 2 | 3 | 3 | 3 |
| Channels multipliers | 1,2,2,2 | 1,2,2,2 | 1,2,3,4 | 1,2,2,2 |
| Heads | 4 | 4 | 4 | 4 |
| Heads Channels | 64 | 64 | 64 | 64 |
| Attention resolution | 16 | 16,8 | 32,16,8 | 32,16,8 |
| Dropout | 0.0 | 0.0 | 0.0 | 0.0 |
| Effective Batch size | 256 | 1024 | 2048 | 1024 |
| GPUs | 2 | 4 | 16 | 16 |
| Epochs | 1000 | 200 | 250 | 100 |
| Iterations | 391k | 250k | 157k | 102k |
| Learning Rate | 4e-4 | 1e-4 | 1e-4 | 1e-4 |
| Learning Rate Scheduler | Polynomial | Polynomial | Constant | Constant |
| Warmup Steps | 45k | 20k | - | - |

objective in the general case. In NFDM, we learn the forward process end-to-end with the reverse process. Thus, reducing the NFDM's objective could lead to the model's collapse, as it is not the correct objective to optimize.

We would also like to emphasize that the NFDM framework, including its objective in eq. (67), represents a generalization of conventional diffusion models, rather than an alternative approach to training. Consequently, if we take an existing forward process from conventional diffusion models and incorporate it into the objective $\mathcal{L}$, we will not create a new model or optimization procedure. Instead, this will align with approaches that focus on likelihood-based training of conventional diffusion models, as explored in [59, 76].

NFDM facilitates the convenient pre-specification of fixed forward processes. Hence, it might be utilized to define a fixed forward process and subsequently train it with the Flow Matching objective. However, this is a topic for future research and is beyond the scope of this work.

## F   Implementation Details

Our evaluation of NFDM and NFBM includes tests on synthetic data, CIFAR-10 [31], two down-sampled ImageNet [11, 67] datasets, and downsampled AFHQ [8]. To maintain consistency with baselines, we employ horizontal flipping as a data augmentation technique in training models on CIFAR-10 and ImageNet [61, 59]. For density estimation of discrete data, uniform dequantization is used (see Appendix F.1).

We parameterize $\hat{\mathbf{x}}_\theta$ (eq. (11)) in the reverse process using a 5-layer perceptrons with 512 neurons in each layer for synthetic data and the U-Net architecture from [12] for images. In all experiments, a 3-layer perceptrons with 64 neurons in each layer is employed to parameterize $g_\varphi$ (eq. (9)).

For the parameterization of $F_\varphi$ (eq. (7)) in NFDM as Gaussian, we employ a neural network identical to that used for $\hat{\mathbf{x}}_\theta$. The only difference is that for $F_\varphi$, we double the output of the last layer to parameterize both $\bar{\mu}_\varphi$ (eq. (52)) and $\bar{\sigma}_\varphi$ (eq. (53)) using the same model (see Appendix B.5).

For parameterizing $F_\varphi$ (eq. (13)) in NFDM, we adopt the same approach but double the input size of the first layer to condition the forward process on both $\mathbf{x}_0$ and $\mathbf{x}_1$ (see Appendix C.1).

To parameterize $F_\varphi$ (eq. (7)) in NFDM as non-Gaussian, we apply the Glow [29] architecture tailored to the corresponding dataset to transform $\varepsilon$ into $\mathbf{z}_t$. To incorporate conditioning, we use a neural network identical to that of $\hat{\mathbf{x}}_\theta$ to generate an embedding $u(\mathbf{x}, t)$. We then add this conditioning to the input of each non-linear transformation in the coupling layers.

To ensure the constraints $g_\varphi \geq 0$ (eq. (9)) and $\bar{\sigma}_\varphi \geq 0$ (eqs. (53) and (58)) are met, we apply the softplus function to the outputs of the neural networks.

The models were trained using the Adam optimizer with the following parameters: $\beta_1 = 0.9$, $\beta_2 = 0.999$, a weight decay of $0.0$, and $\epsilon = 10^{-8}$. The training process was facilitated by a polynomial decay learning rate schedule, which includes a warm-up phase for a predefined number of training steps. During this phase, the learning rate is linearly increased from $10^{-8}$ to a peak value. After reaching the peak learning rate, it is then linearly decreased to $10^{-8}$ by the final training step. The specific hyperparameters are detailed in Table 4. Training was carried out on Tesla V100 GPUs.

### F.1 Dequantization

For reporting the NLL, we employ standard uniform dequantization. The NLL is estimated using an importance-weighted average, given by

$$\log \frac{1}{K} \sum_{k=1}^{K} p_\theta(\mathbf{x} + u_k), \quad \text{where} \quad u_k \sim \mathcal{U}(0, 1), \tag{71}$$

where $\mathbf{x} \in [0, \dots, 255]$.

### F.2 NFBM with Obstacles

In Section 6.3, we demonstrate how the concept of learning NFDM with the restrictions discussed in Section 5 can be adapted to NFBM for learning generative dynamics that avoid obstacles. For this experiment, we propose training the NFBM models with an additional penalty that has higher values at points closer to obstacles. Specifically, we applied the following objective:

$$\mathcal{L}_{\text{obs}} = \mathbb{E}_{u(t)q(\mathbf{z}_t)}[f(\mathbf{z}_t)], \tag{72}$$

where $f(\mathbf{z}_t)$ is a penalty function. In our experiments, we use the probability density of a mixture of Gaussian distributions, truncated at some low values to 0. This truncation is necessary to ensure that the model does not push trajectories infinitely away from the obstacle.

In contrast to $\mathcal{L}_{\text{crv}}$ proposed for NFDM-OT, which penalizes the reverse process influencing the forward process as a result, $\mathcal{L}_{\text{obs}}$ directly penalizes the forward process, thereby influencing the properties of the reverse process.

## G    Additional Results

### G.1    Visualisations of Forward Trajectories of NFDM-OT

In this section, we provide visualizations of the forward trajectories of NFDM-OT, as illustrated in the two-dimensional data distribution experiment shown in Figure 1b. Figure 3 depicts the deterministic trajectories of NFDM-OT. It is easy to see that the forward process has learned some highly curved trajectories. This example highlights several key aspects of NFDM-OT and NFDM more broadly.

Firstly, this example underscores the point discussed in Appendix D.2: while NFDM-OT penalizes the curvature of the reverse process, which is parameterized through the forward process (see Section 3.2), it does not penalize the curvature of the forward process itself.

Secondly, this example demonstrates the importance of learning nonlinear forward processes. In attempting to adapt to the straight-line reverse process, the forward process ends up being highly curved, which in turn aids the reverse process. This emphasizes the importance of the NFDM framework, which facilitates the learning of such nonlinear forward processes that may help to learn better generative dynamics.

Finally, we note that we parameterize the forward process with a Gaussian distributions featuring learnable mean and variance (see Appendix B.5). Even this relatively simple parameterization significantly enhances the flexibility of diffusion models compared to conventional diffusion approaches, enabling the learning of complex forward processes.

### G.2    Additional Results of NFDM and NFDM-OT on Image Datasets

In this section we provide additional results, that directly compare NFDM and NFDM-OT. This results extend Tables 1 and 2. As observed, NFDM without any additional restrictions provides a

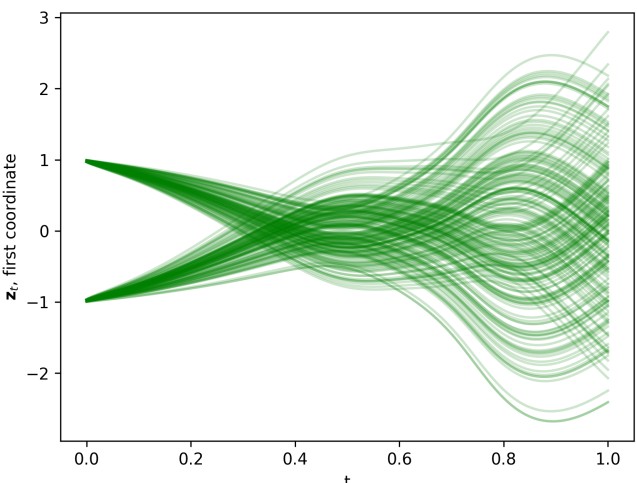

Figure 3: First coordinates of forward deterministic trajectories of NFDM-OT started from points $\mathbf{x} = (-1, -1)$ and $\mathbf{x} = (1, 1)$.

Table 5: Comparison of NFDM with NFDM-OT results on density estimation tasks. We present results in terms of BPD, lower is better.

| Model | CIFAR10 | ImageNet 32 |
|---|---|---|
| NFDM | 2.49 | 3.36 |
| NFDM-OT | **2.62** | **3.45** |

Table 6: Comparison of NFDM with NFDM-OT results on few-step generation. We present results in terms of FID, lower is better.

| | CIFAR-10 | | ImageNet 32 | |
|---|---|---|---|---|
| Model | NFE ↓ | FID ↓ | NFE ↓ | FID ↓ |
| NFDM | 4 | 50.12 | 4 | 57.60 |
| | 12 | 21.88 | 12 | 24.74 |
| NFDM-OT | 4 | 7.76 | 4 | 6.13 |
| | 12 | 5.20 | 12 | 4.11 |

better log-likelihood estimation compared to NFDM-OT. However, as expected, NFDM exhibits much worse sample quality compared to NFDM-OT in a few steps generation. We attribute this property to the higher curvature of NFDM's generative trajectories.

### G.3 Generated Samples

In this section, we present generated samples from the NFDM, NFDM-OT, and NFBM models, trained on various datasets.

**NFDM**: Figure 4 shows samples from the NFDM model with Gaussian $q_\varphi(\mathbf{z}_t|\mathbf{x})$ (see Appendix B.5). We provide samples from NFDM trained on the CIFAR-10, ImageNet 32, and ImageNet 64 datasets.

**NFDM-OT**. Figure 5 displays samples from NFDM trained on ImageNet 64 dataset. We generate these samples by simulating the ODE across 12, 4, and 2 steps.

**NFBM**. Figure 6 demonstrates the generative trajectories of the NFBM model trained on the AFHQ 64 dataset. This model starts from an image of a dog at time step $t = 1$ and simulate the SDE to time step $t = 0$.

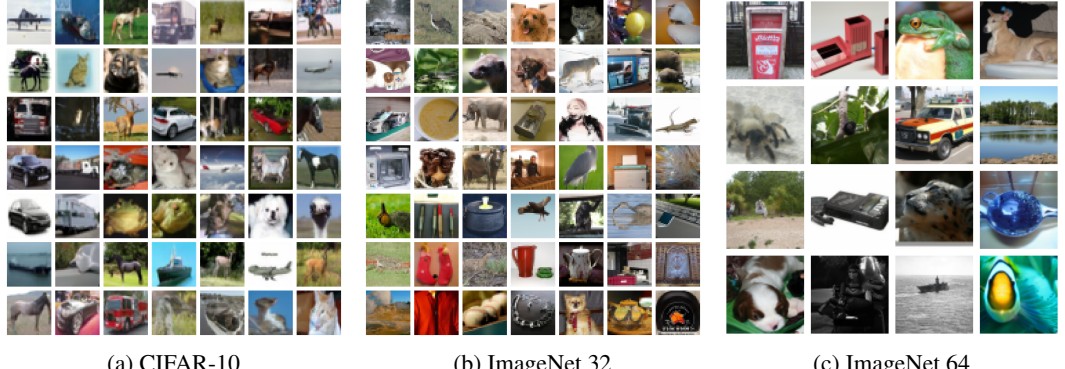

| (a) CIFAR-10 | (b) ImageNet 32 | (c) ImageNet 64 |

Figure 4: Generated samples from NFDM trained on various datasets.

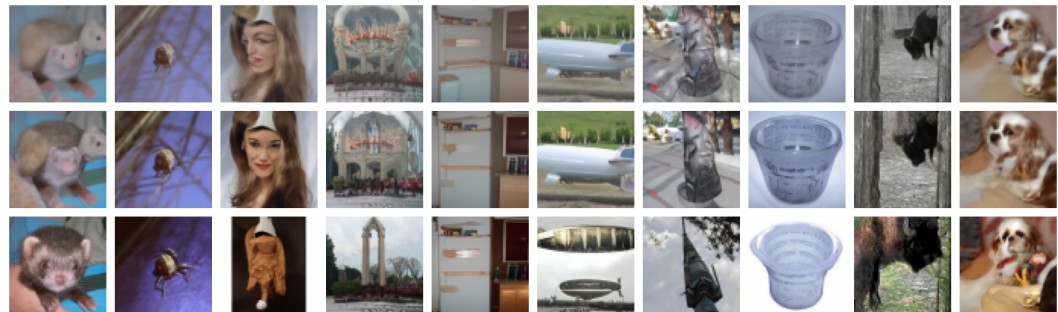

Figure 5: Generated samples from NFDM-OT trained on ImageNet 64. *Top:* NFE = 2; *Middle:* NFE = 4; *Bottom:* NFE = 12.

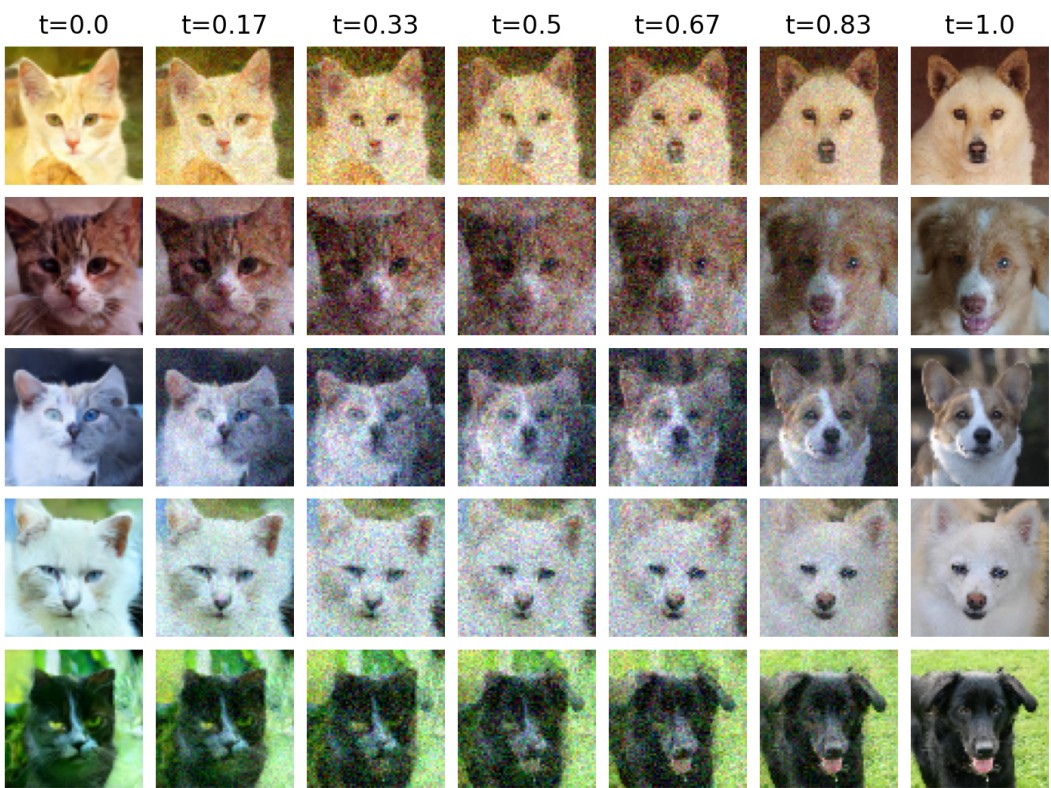

Figure 6: Generative trajectories from NFBM trained on AFHQ 64.

