# OpenReview forum: "Neural Flow Diffusion Models: Learnable Forward Process for Improved Diffusion Modelling"
_NeurIPS.cc/2024/Conference — NeurIPS 2024 poster_

### Official Review · Reviewer_DMsJ · 2024-07-07

**Soundness:** 4
**Presentation:** 4
**Contribution:** 4
**Rating:** 8
**Confidence:** 4

**Summary:**

The paper introduces Neural Flow Diffusion Models (NFDM), which incorporate a learnable forward process to enhance diffusion modeling. Traditional diffusion models rely on a fixed forward process, often complicating the reverse process and resulting in costly inference. NFDM aims to address these challenges by introducing flexibility and improved generative dynamics through a learnable forward process. The paper demonstrates NFDM’s capabilities through experiments on CIFAR-10 and ImageNet datasets, achieving state-of-the-art results in terms of negative log-likelihood. Additionally, NFDM is applied to learn bridges between distributions on the AFHQ dataset, further showcasing its flexibility and utility in different generative modeling tasks.

**Strengths:**

1. The paper introduces the concept of a learnable forward process in diffusion models, which is a novel approach. Traditional diffusion models typically rely on fixed forward processes, and the shift to a learnable process represents a significant departure from established methodologies. This innovation opens up new avenues for research and development in generative modeling.
2. The paper provides a solid theoretical underpinning for the proposed NFDM. The authors thoroughly explain the mathematical principles behind the learnable forward process and its optimization. The experimental validation is robust and extensive. By demonstrating state-of-the-art results on widely recognized benchmarks such as CIFAR-10 and ImageNet, the paper establishes the practical effectiveness of NFDM. The experiments are well-designed and the results are presented with appropriate statistical rigor.
3. The paper is well-written and clearly structured, making it accessible to both experts and those new to the field.
4. By introducing a learnable forward process, the paper makes a significant contribution to the field of generative modeling. This advancement has the potential to influence future research and applications in areas such as image generation.

**Weaknesses:**

1. A previous work Shiftddpms [1] share similar idea with NFDM. It incorporates a learnable forward process (and its corresponding reverse process) to enhance conditional diffusion modeling. The paper doesn't refer to and compare with Shiftddpms.
2. The paper does not extensively discuss the generalization of the results to other settings or the robustness of the results to violations of the assumptions made​. This omission can be critical as it limits the confidence in applying the findings to different datasets or under different conditions than those explicitly tested.

[1] Shiftddpms: Exploring conditional diffusion models by shifting diffusion trajectories

**Questions:**

Please see weaknesses.

**Limitations:**

The authors have adequately described the limitations and potential negative societal impact of their work.

---

> ### Author Rebuttal · Authors · 2024-08-07
>
> We are delighted to see that the reviewer finds our approach novel, the theoretical explanations solid, and the experiments robust, and even recognizes the potential to influence future research. Below, we address the questions and comments raised in the review.
>
> **Weaknesses:**
>
> 1. We thank the reviewer for bringing the Shiftddpms [1] work to our attention, which we had overlooked. While Shiftddpms focuses on conditional generation, it indeed shares similarities with our NFDM, proposing learnable transformations of conditioning. Although we do not discuss conditional generation in this paper, we believe that the NFDM framework could be adapted for conditional generation and this is an interesting avenue for future research.
>
>     We will definitely include a discussion of this work in the revision of the paper if it is accepted.
>
> 2. In this work, we aimed to focus on the introduction of the NFDM itself and demonstrate its capabilities compared to prior work on standard benchmarks used in those works. Indeed, in the current work, we do not discuss topics such as the application of NFDM to domains other than imaging, scaling, conditional generation, or training with non-ELBO objectives. However, we view these as very promising directions for future research.
>
> [1] Shiftddpms: Exploring conditional diffusion models by shifting diffusion trajectories

---

> > ### Comment · Reviewer_DMsJ · 2024-08-11
> >
> > I acknowledge having read the authors' rebuttal. My overall assessment of the paper remains unchanged, and I continue to support my current rating.

---

### Official Review · Reviewer_AXnA · 2024-07-08

**Soundness:** 2
**Presentation:** 3
**Contribution:** 3
**Rating:** 5
**Confidence:** 3

**Summary:**

This work proposes a generic framework to make diffusion model's encoder learnable.

**Strengths:**

The authors propose a neat framework to make the encoding part of the diffusion model trainable. They also explain the differences and relationships between their proposal and related works.

**Weaknesses:**

1. The work starts from the prediction of forward marginal distribution, and derive the conditional ODE and SDE. However, whether these dynamics corresponding to the same marginal distributions need some rigorous reasoning (e.g., Fokker-Planck Equation or continuity equation). For instance, the claim in L132 and L132 should be explained. Please supplement it.

2. The authors claimed: "*conventional diffusion the latent variables are inferred through a pre-specified linear combination of the data point and Gaussian noise. This formulation limits diffusion models in terms of the flexibility of their latent space, and makes learning of the reverse process more challenging...*". However, is there any evidence showing that a diffusion model's linear combination of data and noise may lead to degradation or affect the flexibility of the latent space? Also, if the forward path is simply a pre-specified linear combination of the data point and Gaussian noise, wouldn't it result in a more easily learnable backward process? I suggest the authors provide either theoretical or empirical evidence to support their claim and motivation.

[Minor]
- I appreciate that the authors explained the differences and relationships between their proposal and related works, which is informative. I suggest extending Appendix E.1 to include more comparisons of each method's assumptions, such as the priors used, their final objective functions, and why their parameterizations fall short.

**Questions:**

1. The proposed algorithm suggests learning the encoding and decoding parts together (though alternately). However, the loss function shown in Eq. (12) seems prone to collapsing to a trivial solution, despite the parametrizations potentially mitigating this issue somewhat. Indeed, some literature, such as [1], includes additional regularizers to stabilize the training. Did the authors observe any collapse or instability during training? Any trick to avoid the collapse if it is the case?

2. How $g_\psi$ be determined? How it depends on $\psi$?

3. What is the FID performance of NFDM (with/without curvature regularizer) if the total sampling timestep $T$ is set to the standard diffusion model value of $1000$, compared to DDIM or DDPM? The authors claimed diffusion model's pre-defined encoding may lead to inflexible and complicated latent for reverse process. The training objective is derived in a favor for likelihood computation. However, sample quality should be a more important metric to justify the claim. Also, what does the counterpart of Fig. 2 without curvature regularizer?



[1] S. Lee, B. Kim, and J. C. Ye. Minimizing trajectory curvature of ode-based generative models. arXiv preprint arXiv:2301.12003, 2023.

**Limitations:**

The authors have discussed the limitation that using a learnable forward path may double the training runtime.

---

> ### Author Rebuttal · Authors · 2024-08-07
>
> We are glad the reviewer found NFDM framework as neat. Below, we address the questions and comments raised in the review.
>
> **Weaknesses:**
>
> 1. We define the distribution $q_\phi(z_t|x)$ implicitly through the function $F_\phi(\varepsilon, t, x)$. Under certain assumptions, we can derive an ordinary differential equation (ODE) as shown in Equation 8 by simply differentiating $F_\phi$ with respect to time. It is straightforward to see that integrating this ODE over time produces the same samples as those generated by $F_\phi$. Consequently, the ODE in Equation 8 corresponds to the defined probability density $q_\phi(z_t|x)$.
>
>     The connection between deterministic and stochastic dynamics through the score function, as leveraged in Equations 9 and 10, is a classical result from [1]. Further proofs demonstrating that the dynamics discussed in Equations 8-10 share the same marginals  can be found in [2] (see Appendices A and D.1).
>
>     We will include a more extensive discussion of these derivations in Appendix A in the revised version of the paper, if accepted.
>
> 2. Consider the distribution of latent variables $q(z_t)$ defined by the forward process. Given the data distribution $q(x)$ and the conditional distribution $q(z_t|x)$, we can derive the distribution of latent variables $z_t$ as follows: $q(z_t) = \int q(x_t|x) q(x) dx$. Therefore, if the forward process that defines $q(z_t|x)$ is fixed, $q(z_t)$ is also implicitly predefined and fixed. At the same time, the generative process must adhere to all intermediate distributions $q(z_t)$ starting from $t=1$. This is why a fixed forward process limits the flexibility of the latent space.
>
>     From a theoretical perspective, conventional diffusion with a fixed forward process that only injects Gaussian noise is a model that in the ideal scenario works for arbitrary distributions. However, conventional diffusion can be suboptimal, potentially leading to distributions of latent variables that are difficult to learn or highly curved trajectories (Fig. 1) that are expensive to integrate. A learnable forward process could simplify the target for the generative process, thereby improving the alignment of forward and generative processes and better capturing the data distribution as a result.
>
>     We provide empirical evidence that learning the forward process enhances the performance of the generative process. A theoretical motivation for learning the forward process is also detailed in Appendix B.1.
>
>     Additionally, the flexibility of the NFDM framework allows us to learn not just any generative process, but one with desired properties such as straight-line deterministic trajectories (see Section 5), which is unachievable with conventional diffusion that predetermines the distributions of the latent variables.
>
> 3. We are pleased that the reviewer appreciated our discussion of connections with related works. We will elaborate on the differences with these approaches in more detail in the revision of the paper, if accepted. We note, however, that in most cases, the parameterizations of NFDM discussed in Appendix E.1 lead to the same objectives as proposed in related works.
>
> **Questions:**
>
> 1. In our experiments, we intentionally parameterize the forward process to ensure that $q_\phi(z_0|x) \approx \delta(x - z_0)$ and $q_\phi(z_1|x) \approx p(z_0)$ (see Appendix B.5). In this way, the model can not collapse by design. For instance, in Eq. 12, $\tilde{f}_\phi(z_t, t, x)$ can not always equal $0$. Furthermore, the NFDM objective is also a variational bound on the log-likelihood (Appendix A.1).
>
>     In practice we did not observe any instabilities in the training procedure and did not apply any techniques to stabilize training.
>
> 2. We describe the parameterization of $g_\phi$ in Appendix F. In all our experiments, $g_\phi$ is parameterized using a 3-layer perceptron with 64 neurons in each layer.
> 3. When sampling with an adaptive ODE solver from models trained on the CIFAR10 dataset, we achieve an FID score of 4.22 for NFDM (with an average of 82 NFE) and 4.85 for NFDM-OT (with 45 NFE on average). For reference, a DDPM trained with the ELBO objective has an FID score of 13.51 (with 1000 NFE). We plan to calculate the FID metrics for the ImageNet dataset by the end of the discussion period.
>
>     Some modifications of diffusion models trained with better parameterizations and different objectives can achieve lower FID scores, but they tend to show worse log-likelihood estimates. This phenomenon is a well-known attribute of diffusion models. We believe that with improved parameterization and hyperparameter- and objective tuning (which was done for conventional diffusion), the flexibility of NFDM could lead to even better performance in terms of FID score, but we leave this for future research.
>
>     Although we have provided experimental results on image datasets as a standard benchmark, we introduce NFDM as a versatile generative modeling framework. In this context, likelihood emerges as the most commonly employed metric in probabilistic modeling across various fields, and NFDM achieves state-of-the-art results. While the FID score is a popular metric for images, it does not always correlate with image quality [3]. In contrast, likelihood is based on robust theoretical principles. We will include a table with these FID scores in the appendix.
>
>
> We trust that the clarifications further explanations of motivation, technical details and other will enhance your support for our paper's acceptance!
>
> [1] B. D. Anderson. Reverse-time diffusion equation models. Stochastic Processes and their Applications, 12(3):313–326, 1982.
>
> [2] Song, Yang, et al. "Score-based generative modeling through stochastic differential equations.”
>
> [3] Stein, George, et al. "Exposing flaws of generative model evaluation metrics and their unfair treatment of diffusion models."

---

> ### Comment · Reviewer_AXnA · 2024-08-12
> **Thanks for clarification and replies.**
>
> I appreciate the reviewers' clarification and have decided to increase my score.

---

### Official Review · Reviewer_1ZLG · 2024-07-11

**Soundness:** 3
**Presentation:** 2
**Contribution:** 3
**Rating:** 7
**Confidence:** 3

**Summary:**

The authors propose a variant of a diffusion model in which the forward process is parameterized by a neural network that maps from data, noise, and a timestep to the noised data. They show how, if this neural network has certain properties, the SDE and ODEs of the forward and reverse processes can be simulated. They demonstrate good density estimation results and the ability to penalize the learned forward process to encourage the straightness of ODE trajectories.

**Strengths:**

- An elegant proposed framework for generalizing diffusion
- I think that learning modifications to the forward process is a potentially very impactful research direction and the ability to express constraints on it opens up new avenues for exploration
- State-of-the-art density estimation results

**Weaknesses:**

- Exposition could be clearer. Understanding all of the notation in Section 3 in particular took a while. I understand that some of this complexity might be irreducible, but it could be helpful to e.g. have a diagram showing relationships between e.g. the different variaions of , what
learns to predict, the direction stepped in during the reverse ODE, etc.
- It would be interesting to see comparisons between NFDM and NFDM-OT included in the paper. E.g. if both of them could be included in each of the results tables. It would be very interesting to see how they compare FID and NLL, and also how the learned trajectories of NFDM compare to those of Fig 1a.

**Questions:**

See weaknesses

**Limitations:**

Adequately addressed

---

> ### Author Rebuttal · Authors · 2024-08-07
>
> We are pleased that the reviewer appreciates the elegance of our framework and recognizes the potential impact of our research direction. In response to the comments and questions raised, we provide the following clarifications and commitments:
>
> **Weaknesses:**
>
> 1. We acknowledge the lack of visual explanations in our paper. Due to space constraints in the main manuscript, we unfortunately did not include them. In the camera-ready version, if accepted, we plan to include a series of figures in the main paper or the appendix. Specifically, we intend to add a diagram that visually explains how the $F_\varphi$ function defines the conditional distributions $q_\varphi(\mathbf{z}_t|\mathbf{x})$, and illustrates the connection between ODEs and SDEs corresponding to the forward process.
>
>     Additionally, in our global response, we have provided visualizations of generated samples from the NFDM and NFBM models trained on various datasets.
>
> 2. We have provided a comparison between NFDM and NFDM-OT in Tables 5 and 6 in the appendix. Compared to NFDM, NFDM-OT trades off worse density estimation performance (lower log-likelihood) for better sample quality (lower FID) in the few-step generation mode.
>
>     In the same 2D experiment illustrated in Figure 1, NFDM exhibits slightly less curved trajectories compared to conventional diffusion. We will include additional illustrations in the camera-ready version, if accepted.
>
>
> We believe that the additional visualizations and comparisons will strengthen your support for acceptance. We are also happy to address any further concerns or questions the reviewers may have.

---

> > ### Comment · Reviewer_1ZLG · 2024-08-13
> >
> > Thank you for the response. It addressed my concerns and I've raised my score to a 7.

---

### Official Review · Reviewer_xgai · 2024-07-12

**Soundness:** 3
**Presentation:** 3
**Contribution:** 2
**Rating:** 6
**Confidence:** 4

**Summary:**

The paper introduces Neural Flow Diffusion Models (NFDM), a novel framework that enhances diffusion models by supporting a broader range of forward processes beyond the standard Gaussian. The proposed parameterization technique facilitates the learning of the forward process and minimizes a variational upper bound on the negative log-likelihood in an end-to-end, simulation-free manner. Experimental results on datasets like CIFAR-10, ImageNet 32, and 64 demonstrate state-of-the-art performance in terms of log-likelihood and generation quality.

**Strengths:**

- The paper is well-written and easy to follow.
- The problem is conceptually well-motivated and interesting.
- The introduced framework is clearly explained with all the necessary details.
- Development of a new optimization framework preserving simulation-free.
- Demonstration of state-of-the-art results on CIFAR-10, ImageNet 32, and 64.
- Application of NFDM and NFBM with specific properties and improved sampling speed.

**Weaknesses:**

1. Previous works are mentioned but the introduced NFDM framework is not properly placed in the context of the related works. Therefore, it is difficult to fully assess the novelty and the contributions.
1. It is unclear how the introduced method would scale to larger datasets and image resolutions and what would be the benefit.
1. The non-Gaussian parameterization does not show substantial improvement over the Gaussian parameterization, raising questions about its necessity.

**Questions:**

1. Can the authors provide a clear statement of their contributions and methods within the context of related works that also aim to extend the diffusion framework to include a learnable and potentially non-linear forward process (e.g. [2])? Specifically, it would be helpful to highlight the advantages and disadvantages of these methods compared to theirs, from both methodological and computational perspectives.
2. How do the authors ensure that the function $F$ is smooth between $z_0$​ and $z_1$​ (Line 120)?
3. Can the authors clarify how the linear in $\epsilon$ parameterization of $F_\varphi$ relates to the stochastic interpolant framework in  [1].  Such comparison is already outlined in Appendix E where the authors point out the different optimization problems (min-min vs min-max) and state "*Stochastic Interpolants necessitate learning two separate functions for the reverse process: the velocity field and the score function*".  Could the authors clarify the latter statement? It seems that also in the proposed NFDM framework one would need to learn both the $F_\varphi$ (which moreover needs to be derived wrt to time and invertible wrt to $\varepsilon$) and the prediction network $\hat{x}_\theta$ .
4. For clarity of exposition, it would be beneficial to better introduce the function $\hat{x}_\theta$​, particularly explaining the difference with the score network in conventional diffusion.
5. Why is the framework still considered simulation-free despite requiring a prediction network $\hat{x}_\theta$​? Can the authors explain why this step does not imply that the model, parameterized by $\theta$, implicitly learns to simulate the process?
6. In Table 1, there appears to be no significant improvement when NFDM uses a Gaussian parameterization compared to a non-Gaussian one. Could the authors comment on this result?
7. In App B.3, the authors also mention that a non-Gaussian parameterization is not scalable for high-dimensional cases.  Have the authors observed any case where a non-Gaussian parameterization leads to an improvement?
8. Can the authors offer a more detailed comparison of their _Restricted NFDM_ compared to previous works that also constrain the generative process to have straight trajectories? Specifically, what are the computational advantages/disadvantages?
9. Can the authors better discuss the computational cost of NFDM compared to previous approaches? If I am not mistaken I only see a mention in the conclusion "*this leads to approximately 2.2 times longer optimization iteration of NFDM compared to conventional diffusion models*" but it is not clear to which NFDM parameterisation this estimate refers.  Moreover, while Table 2 depicts a clear advantage of NFDM in terms of NFE, it would be useful to analyze these numbers in the context of the computational complexities of the respective methods.

[1]  Albergo, Michael S., and Eric Vanden-Eijnden. "Building normalizing flows with stochastic interpolants." arXiv preprint arXiv:2209.15571 (2022).


[2] Kim, Dongjun, et al. "Maximum likelihood training of implicit nonlinear diffusion model." _Advances in neural information processing systems_ 35 (2022): 32270-32284.

**Limitations:**

The limitations are addressed in the paper.

---

> ### Author Rebuttal · Authors · 2024-08-07
>
> We appreciate the reviewer's positive feedback regarding the clarity and presentation of our paper. Below, we address the questions and comments raised in the review.
>
> **Weaknesses:**
>
> 1. Please refer to the answer to Question 1 below.
> 2. Unfortunately, due to limited computational resources, we were not able to conduct experiments on a significantly larger scale than those on ImageNet 64. Consequently, we cannot definitively ascertain the outcomes in a large-scale setting.
>
>     Nevertheless, we would like to highlight that NFDM is a versatile framework that includes conventional diffusion as a special case which has been shown to scale. Furthermore, CIFAR-10 and ImageNet are standard benchmarks in the diffusion literature. Although we lack large-scale experiments to support our position fully, we see no reasons that would impede the scaling of NFDM.
>
> 3. Please refer to the answer to Question 6 below.
>
> **Questions:**
>
> 1. The main contribution of our work is the neural flow diffusion model, introduced as a general framework.  It enables the construction of diffusion models with a broad spectrum of forward processes, including *non-linear*, *time-dependent*, and *learnable processes*. We show how to efficiently parameterize and train diffusion models with such forward processes, while retaining key properties of diffusion models, such as access to stochastic and deterministic generative dynamics as well as simulation-free training.
>
>     In Appendix E, we discuss in more detail the connections between NFDM and related works that introduce learnable parameters into the forward process. Most existing prior work can be considered special cases of NFDM. For instance, to parameterize the forward process in [1] using NFDM, one could set $F_\phi(\varepsilon, t, x) = \alpha_t E_\phi(x) + \sigma_t \varepsilon$, where $E_\phi(x)$ is an invertible transformation.
>
>     We emphasize that NFDM is not merely an alternative approach, but a generalization of prior diffusion models. With a simple parameterization such as $F_\phi(\varepsilon, t, x) = \alpha_t x + \sigma_t \varepsilon$, NFDM directly corresponds to DDPM and introduces no additional computational overhead. The computational cost of NFDM depends on the complexity of the forward process' parameterization, which lets the user interpolate between standard linear transformations to highly complex nonlinear ones.
>
>     To the best of our knowledge, NFDM provides the most general diffusion modelling framework that allows learning the forward process while keeping the training procedure simulation-free. Besides achieving state-of-the-art log-likelihood modelling results, the flexibility of NFDM enables us to learn generative processes with specific properties like NFDM-OT, for which it is crucial that the forward process has a complex non-linear time-dependent forward processes (see Appendix G.1).
>
> 2. To ensure that $F_\phi$ is smooth in time, we chose a parameterization that is continuous and differentiable. These requirements are standard for the neural networks and are not difficult to meet. We discuss the parameterization of $F_\phi$ in more detail in Appendix B.5 and implementation details in Appendix F.
> 3. The linear parameterization in $\varepsilon$ of $F_\phi$ is as follows: $F_\phi(\varepsilon, t, x) = \mu_\phi(x,t) + \sigma_\phi(x,t) \varepsilon$, where $\mu_\phi$ and $\sigma_\phi$ are learnable functions. In terms of NFDM, we may describe the forward process from [2] as $F_\phi(\varepsilon, t, x) = (1 - t) x + t \varepsilon$. In [3], the authors discuss other variations of linear combinations of $x$ and $\varepsilon$, which can also be easily expressed within the NFDM framework. A key distinction between NFDM and [2, 3] is that the Stochastic Interpolants framework does not propose a method to learn the forward process. Their derivations rely on a fixed forward process, while NFDM allows for learnable parameters in the forward process.
>
>     Indeed, when we parameterize the forward process of NFDM with learnable parameters, we need to learn both the function $F_\phi$ and the predictor $x_\theta$. In Appendix E, we referred to the fact that in [3], with a fixed forward process, two different functions need to be learned for inference, one for the velocity and one for the score. In contrast in NFDM, if we fix the forward process (i.e., fix $F$) like in [3], we only need to learn the predictor $\hat{x}_\theta$.
>
> [1] Kim, Dongjun, et al. "Maximum likelihood training of implicit nonlinear diffusion model.”
>
> [2] Albergo, Michael S., and Eric Vanden-Eijnden. "Building normalizing flows with stochastic interpolants.”
>
> [3] Albergo, Michael S., Nicholas M. Boffi, and Eric Vanden-Eijnden. "Stochastic interpolants: A unifying framework for flows and diffusions.”

---

> > ### Author Response · Authors · 2024-08-07
> >
> > 4. In conventional diffusion, where the forward process is parameterized as $F(\varepsilon, t, x) = \alpha_t x + \sigma_t \varepsilon$, there is a straightforward connection between the score predictor and the predictor of the data point: $s_\theta(z_t, t) = \frac{\alpha_t x_\theta(z_t, t) - z_t}{\sigma_t^2}$ (see [4]). Thus, during the training procedure, we can learn to predict either the data point predictor or the score function. Then, after training, any of these functions can be solved for from the other. From a theoretical standpoint, all these parameterizations are equivalent.
> >
> >     When the forward process of NFDM is parameterized linearly as discussed above, NFDM exactly corresponds to conventional diffusion, and the function $x_\theta$ maintains the same relationship with the score function. However, in the general case, when $F_\phi$ is an arbitrary function, these connections between $s_\theta$ and $x_\theta$ do not exist. In our work, the parameterization of the reverse process through the function $x_\theta$ is a design choice. As discussed, $x_\theta$ retains relationships with the score function in the linear case. However, in the general case, we do not expect it to possess any special properties. We will include a discussion on this in the camera-ready, if accepted.
> >
> > 5. In the literature on diffusion models, the term "simulation-free" is most often taken to mean that during training the method does not require numerical simulations of the SDE or the integration of any dynamics at the training stage. For instance, approaches like [5], or others based on Schrödinger Bridge theory, require numerical simulation of an SDE that describes the forward process to sample $z_t \sim q(z_t|x)$. Such requirements make the training procedure significantly more expensive and are difficult to scale.
> >
> >     In contrast, the NFDM framework allows for direct sampling of $z_t$ by inferring the function $F_\phi$, which includes no numerical simulations. We also do not need to simulate the generative dynamics during training, making the NFDM framework simulation-free.
> >
> > 6. As previously mentioned, the main contribution of our work is a comprehensive framework that supports a wide range of forward processes. The primary objective of the experiments with non-Gaussian parameterization of the forward process is not to surpass the Gaussian parameterization but to demonstrate the flexibility of the NFDM framework and its capability to handle even such complex parameterizations.
> >
> >     We believe there are two main reasons why the non-Gaussian parameterization of the forward process does not show significant improvement compared to Gaussian parameterization. First, we have only evaluated our approach on image datasets, where Gaussian noise appears to naturally fit well. Secondly, we chose a relatively simple parameterization of the non-Gaussian forward process, which may be suboptimal. It is possible that in different domains or with a more powerful parameterization, non-Gaussian parameterization may prove more beneficial, but we leave this for future research. In this work, we aimed to demonstrate the functionality of the NFDM framework.
> >
> >     Additionally, we would like to emphasize that even the Gaussian parameterization of NFDM with data-dependent variance is a generalization of existing diffusion models and leads to state-of-the-art log-likelihood estimation results.
> >
> > 7. Indeed, in Appendix B.3, we discuss that *a naive parameterization* of $F_\phi$ with arbitrary non-Gaussian functions is not scalable (density evaluation can be O(dim^3)). However, if we parameterize $F_\phi$ with functions that provide easy access to the log-determinant of the Jacobian of transformation, we can apply this parameterization even in the high-dimensional case. This is why, in our experiments on images with non-Gaussian parameterization of the forward process, we use a Glow neural network architecture. This architecture is scalable. Our experiments in Table 1 demonstrate that this non-Gaussian parameterization can lead to better log-likelihood estimation results, achieving state-of-the-art *on both* CIFAR-10 and ImageNet.
> >
> > [4] Kingma, Diederik, and Ruiqi Gao. "Understanding diffusion objectives as the elbo with simple data augmentation.”
> >
> > [5] Zhang, Qinsheng, and Yongxin Chen. "Diffusion normalizing flow.”

---

> > > ### Author Response · Authors · 2024-08-07
> > >
> > > 8. In the paper, we consider two related works that also propose approaches to learn straighter generative trajectories: [6] and [7].
> > >
> > >     The approach in [6] may be seen as a special case of NFDM (see the last paragraph in Appendix E.1). As soon as the authors in [6] propose to learn an additional function to parameterize the forward process, the training complexity of their method is comparable with NFDM. However, in NFDM-OT, we also need to calculate the curvature penalty (eq. 16), which makes the training more expensive compared to [6].
> > >
> > >     In [7], the authors proposed constructing the forward process with optimal transport data-noise couplings. The computational disadvantage of this approach is the necessity to find an optimal coupling for each mini-batch, which may be costly for large batch sizes and some complicated metrics (e.g., distance between 3D structures).
> > >
> > >     In both [6] and [7], the inference works like in conventional diffusion models. However, due to the chosen parameterization, the inference of NFDM is more computationally expensive. In our experiments, instead of predicting the velocity field directly, we predict the data point $x$ and then substitute this prediction into the forward process. Therefore, we need to infer two functions: $x_\theta$ and $F_\phi$.
> > >
> > >     The primary goal of NFDM-OT is to demonstrate the NFDM framework flexibility and its capability to learn generative dynamics with desired properties. In contrast to [6, 7], which only allows minimisation of trajectories curvature, NFDM allows learning generative dynamics with a variety of different properties like we demonstrate in Figure 2. Nevertheless, NFDM-OT is able to efficiently learn straight-line generative trajectories and demonstrates better performance compared to prior works as illustrated in Table 2.
> > >
> > > 9. In our experiments, we observe a similar slowdown for both Gaussian and non-Gaussian parameterizations of NFDM. The training iteration of NFDM takes approximately 2 times longer compared to conventional diffusion models that don’t parameterize the forward process. This happens because we use neural networks of similar sizes to parameterize both the forward and the reverse processes.
> > >
> > >     Similarly, due to the parameterization of the generative process which uses the forward process, one inference iteration also takes approximately 2 times longer compared to the inference of conventional diffusion.
> > >
> > >     We would like to emphasize that such properties of the models are consequences of these design choices. For simplicity of the experimental setup, we use neural networks of the same size to parameterize the forward process, which might be redundant. Additionally, it’s possible to choose an alternative parameterization of the reverse process that wouldn’t rely on the forward process for inference. That’s why we think that the comparison under the same number of function evaluations is more important.
> > >
> > >     Nevertheless, even with this current parameterization, NFDM-OT demonstrates better performance compared to prior work under the same computational budget (if we double the NFE in Table 2).
> > >
> > > We hope that the clarifications provided, highlighting the novelty and strong performance of our work, along with further explanations of technical details will enhance your support for our paper's acceptance!
> > >
> > > [6] Lee, Sangyun, Beomsu Kim, and Jong Chul Ye. "Minimizing trajectory curvature of ode-based generative models.”
> > >
> > > [7] Pooladian, Aram-Alexandre, et al. "Multisample flow matching: Straightening flows with minibatch couplings.”

---

> > > > ### Comment · Reviewer_xgai · 2024-08-12
> > > >
> > > > I appreciate the time and effort the authors invested in addressing the concerns I raised. Together with your clarifications, I consider your work to be a relevant contribution to the field, and therefore, I have decided to raise my score.

---

### Author Rebuttal · Authors · 2024-08-07

We thank the reviewers for their valuable feedback, which helps us to improve the paper. Below, we have responded individually to the specific comments and questions from each reviewer.

Here we would like to provide some visualizations of samples generated with the NFDM and NFBM models trained on different datasets. Please see the attached PDF file.

We hope that the additional visualizations, along with the clarifications and explanations provided below, will strengthen your support for the acceptance of our paper!

---

### Comment · Area_Chair_eTKV · 2024-08-12
**Discussion**

Please participate to the discussion.

---

### Author Response · Authors · 2024-08-14

We would like to thank reviewers for their time and valuable contributions and discussion, which enhance the quality of our work. We are grateful for the reviewers' recognition of the NFDM framework as a “significant contribution”, “elegant”, “neat”, “novel” and “clearly explained”. We also appreciate the acknowledgment of our experiments as "robust and extensive," and the recognition of the potential of this research direction to be "impactful" and "influence future research." We are especially thankful for the reviewers’ unanimous recommendations.

Here we summarise the main contributions of NFDM and the discussion:

- We present NFDM, a novel general framework for diffusion models, which allows non-linear, time-dependent and learnable forward processes.
- We present novel theoretical results, where we derive a simulation-free objective for NFDM, which is crucial for efficient training. We also derive ordinary and stochastic differential equations for sampling.
- NFDM subsumes a large portion of existing approaches as special cases, enabling the free specification of the forward process (Section 7 and Appendix E). Moreover, it extends these much further, offering additional flexibility by introducing learnable parameters.
- NFDM, with a simple parameterization of learnable transformations and without hyperparameter tuning, consistently and significantly outperforms baselines in density estimation across a range experiments (Table 1).
- With a simple modification, our framework can also facilitate building bridges between two different distributions, NFBM.
- NFDM opens a prospect for building new generative dynamics with specific properties, which conventional diffusion models like DDPM and DDIM are incapable of.
- NFDM with additional penalty on curvature of deterministic generative trajectories (NFDM-OT), enables the learning of straight-line generative dynamics. This advancement permits the generation of samples in significantly fewer steps, outperforming baselines in few-step generation (Table 2).
- NFDMs does not require any additional techniques to ensure stable training.

We trust that the insights gained from our discussion will bolster your support for acceptance!

---

### Decision · Program_Chairs · 2024-09-25

**Decision:**

Accept (poster)

**Comment:**

This paper proposes Neural Flow Diffusion Models (NFDM), a framework that allows one to introduce a learnable forward process into diffusion models, enabling more flexible forward processes beyond the standard Gaussian.

Reviewers recognize the paper's significant contribution, clear presentation, strong theoretical foundation, and impressive empirical performance on standard benchmarks. I concur with them that the paper presents an interesting methodology and I thus recommend the paper to be accepted. However, when preparing the final version of the paper, the authors should address carefully the remaining concerns of the reviewers. To strengthen the paper, the authors should provide a more thorough comparison with prior work (shiftddpms in particular), conduct more extensive experiments, and clarify the points of confusion regarding the framework's details. Addressing these concerns will significantly enhance the paper's impact.